# CAUSAL EFFECT ESTIMATION WITH MIXED LATENT CONFOUNDERS AND POST-TREATMENT VARIABLES

## ABSTRACT

Causal inference from observational data has attracted considerable attention in recent years. One main obstacle is the handling of confounders. As the direct measure of confounders may not always be feasible, recent methods seek to address the confounding bias with proxy variables, which are covariates researchers postulate to be conducive to the inference of latent confounders. However, observed covariates may scramble both latent confounders and latent post-treatment variables in observational study, where existing methods risk biasing the estimation by unintentionally controlling for variables affected by the treatment. In this paper, we systematically investigate the bias due to latent post-treatment variables, i.e., *latent post-treatment bias*, in causal effect estimation. We first derive the bias of existing methods when selected proxies scramble both latent confounders and latent post-treatment variables, which we demonstrate can be arbitrarily bad. We then propose a novel Confounder-identifiable VAE (CiVAE) to address the bias. CiVAE is built upon the assumption that the prior of the latent variables belongs to a general exponential family with at least one invertible sufficient statistic in the factorized part. Based on this, we show that latent confounders and latent post-treatment variables can be individually identified up to simple bijective transformations. Finally, we prove that the true causal effects can be unbiasedly estimated with the transformed confounders inferred by CiVAE. Experiments on both simulated and real-world datasets demonstrate that CiVAE is significantly more robust to latent post-treatment bias than existing methods for causal effects estimation.

## 1 INTRODUCTION

Causal inference, which seeks to draw conclusions about cause-and-effect relationships among variables of interest, has gained increasing prominence in various fields, such as social science, economics, and public health (Glass et al., 2013; Johansson et al., 2016; Prosperi et al., 2020). Traditional methods rely on randomized control trials (RCT) to draw valid causal conclusions from experimentation (Cook et al., 2002). Recently, more attention has been dedicated toward causal inference from observational datasets, which contain samples with passively observed past treatment, the associated outcome, and possibly features, and in which researchers have no control over the treatment assignment mechanism (Shalit et al., 2017; Shi et al., 2019; Wager & Athey, 2018).

One main obstacle to inferring causal relations from observational data is confounding bias, which occurs when past treatments were determined by variables that causally influence the outcome, i.e., confounders. In such cases, the difference in the average outcome between the treatment group and the non-treatment group cannot be attributed solely to the treatment, but may also be due to the systematic difference of samples in the two groups (Mickey & Greenland, 1989). If the confounders can be observed, a simple strategy to address such a bias is to control them via methods such as covariate adjustment (Pocock et al., 2002) or propensity score re-weighting (Li et al., 2018). However, confounders are not always measurable (Kuroki & Pearl, 2014). Therefore, recent methods seek to adjust for the influence of confounders based on their noisy proxies, which are generally covariates researchers postulated to be conducive for the inference of confounders (Miao et al., 2018; Yao et al., 2018; Madras et al., 2019). One exemplar work from this strain is the causal effect variational auto-encoder (CEVAE) (Louizos et al., 2017) (Fig. 1-(a)), which has demonstrated that confounding bias can be mitigated by controlling latent variables inferred from proxies of confounders.

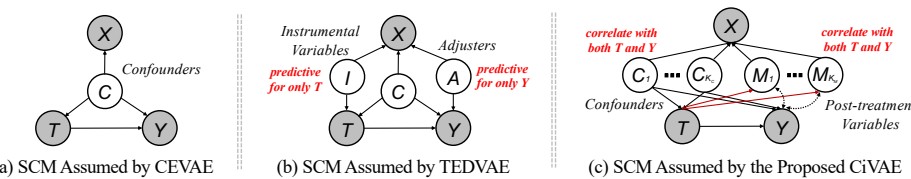

Figure 1: Structural causal model (SCM) assumed by CEVAE, TEDVAE, and CiVAE.

Although proxy-of-confounder-based methods have achieved substantial progress, we argue that these algorithms may risk controlling latent post-treatment variables (i.e., variables causally affected by the treatment) scrambled in the proxy variables, where **post-treatment bias** may be unintentionally introduced in the estimated treatment effect. Here, we note that the negative effects of controlling post-treatment variables have been investigated in prior research (Acharya et al., 2016; Elwert & Winship, 2014; King & Zeng, 2006). For example, Montgomery et al. (2018) found that more than 50% of the papers published in top journals of politics inadvertently control post-treatment variables in the experimental setting, although researchers have complete control over the treatment assignment mechanism and the covariates to control for. On this basis, we postulate that post-treatment bias could be even worse for proxy-based methods in the setting of observational study, as when treatments are passively recorded, it is difficult to determine which variables causally influence the treatment and which variables are influenced by it (as both confounders and post-treatment variables are correlated with the treatment and the outcome). In addition, the post-treatment variables can be latent, which may be scrambled into the observed covariates together with the latent confounders.

Consider the following real-world example that researchers from the Company[1] have encountered when estimating the average causal effects of *switching a job from onsite to online mode* to *the statistics of the applicants* (e.g., average age, gender/geographical diversity, etc.). In this case, the Company collected a dataset of two groups of online (i.e., the treatment group) and onsite jobs (i.e., the non-treatment group), where for each job, the statistics of the applicants (i.e., the average age) are calculated as the outcome. Clearly, the seniority of the job is a confounder between the treatment and the outcome, as less senior jobs (e.g., internships) are more likely to permit online work, and applicants for these jobs tend to be younger on average. The seniority of a job can be difficult to measure. Therefore, the required skills of the job, which the recruiter must provide when publishing a job Ad in the Company, can be used as the proxy of the confounder "seniority". However, **a caveat** is that, switching to an online working mode may also alter the required skills of a job, thereby affecting the qualification of the applicants (where these altered skills are mediators). Consequently, directly using the required skills as the proxy of the confounder "seniority" could unintentionally control latent mediators, which introduces post-treatment bias in the causal effect estimation results.

Addressing the **latent post-treatment bias** faces multi-faceted challenges. First, there lacks a theoretical formulation of the bias when the selected proxies scramble latent post-treatment variables for proxy-of-confounder-based methods; the trade-off between deconfounding and introducing new post-treatment bias is not clear. In addition, it is difficult to distinguish confounders and post-treatment variables in the latent space. Existing covariate disentanglement-based methods, e.g., TEDVAE (Zhang et al., 2021), mainly focus on an easier task of disentangling latent confounders with latent adjusters and instrumental variables. This can be achieved by using their different predictive abilities w.r.t. the treatment and outcome (see Fig. 1-(b)). However, since latent confounders and post-treatment variables correlate with both the treatment and outcome, the two cannot be disentangled by these methods. One solution is to assume the proxy of latent post-treatment variables can be observed, from which post-treatment variables can be inferred and disentangled from the latent confounders. However, this assumption is **too strong**, as in the previous online/onsite job case, we can never know which skills are causally influenced by the work mode. Finally, even if latent confounders can be distinguished, since general latent variable models have no identifiability guarantee (Khemakhem et al., 2020), it is unclear whether controlling the inferred latent variables, which may be arbitrary transformations of the true confounders, can provide unbiased estimations of the causal effects.

To address the aforementioned challenges, we provide a systematic investigation of the latent post-treatment bias in causal inference. We first analyze the behavior of existing proxy-based causal inference methods when the selected proxies scramble both latent confounders and post-treatment variables, where we demonstrate that the estimated average causal effects can be arbitrarily biased.

---

[1]Anonymized due to double-blind review policy.

We then propose the Confounder-identifiable VAE (CiVAE) to address such biases. Specifically, we show that based on a mild assumption that the prior distribution of latent variables (i.e., the latent confounders and post-treatment variables) belongs to a general exponential family with at least one invertible sufficient statistic in the factorized part, latent confounders and latent post-treatment variables can be *individually* identified up to *simple bijective transformations*. In addition, based on the causal relations among confounders, mediators, and treatment, we further demonstrate that the inferred confounders (which are actually transformed proxies of the true confounders) could be properly distinguished from the inferred latent post-treatment variables with pair-wise conditional independence tests. Finally, we prove that the true causal effects can be unbiasedly estimated based on transformed confounders inferred by CiVAE. Experiments on both simulated and real-world datasets demonstrate that CiVAE shows more robustness to latent post-treatment bias than existing methods.

## 2 PROBLEM FORMULATION AND ANALYSIS

### 2.1 PROBLEM FORMULATION

Throughout this paper, we assume the causal model in Fig. 1-(c), where the dashed lines denote indeterminate causal mechanisms that might vary in different cases. We use a binary random variable $T$ to denote the treatment, a random vector $\boldsymbol{X} \in \mathbb{R}^{K_X}$ to denote the observed covariates, and a random scalar $Y \in \mathbb{R}$ to denote the outcome. Furthermore, observed covariates $X$ are assumed to be generated from $K_C$ independent latent confounders $\boldsymbol{C} \triangleq [C_1, C_2..., C_{K_C}]$ and $K_M$ latent post-treatment variables $\boldsymbol{M} \triangleq [M_1, M_2..., M_{K_M}]$ under the causal influence of treatment $T$. We use the random vector $\boldsymbol{Z} \triangleq [\boldsymbol{C} || \boldsymbol{M}] \in \mathbb{R}^{K_Z = K_C + K_M}$ to denote all latent factors. **Our aim** is to estimate the average causal effects of treatment $T$ on outcome $Y$ with auxiliary confounder information in $\boldsymbol{X}$, where the estimation should be devoid of both confounding bias and post-treatment bias.

### 2.2 ANALYSIS OF LATENT POST-TREATMENT BIAS

#### 2.2.1 PRELIMINARIES AND ASSUMPTIONS

To achieve such a purpose, we first formally define the (conditional) average treatment effects (C/ATE) when covariates $\boldsymbol{X}$ scramble both latent confounders $\boldsymbol{C}$ and post-treatment variables $\boldsymbol{M}$. We then define the post-treatment bias when covariates $\boldsymbol{X}$ are used directly as the proxy of confounders. To facilitate the analysis, we make the following assumption regarding the causal generative process.

**Assumption 1.** *(**Noisy-Injectivity**). We assume $\boldsymbol{X} = f(\boldsymbol{C}, \boldsymbol{M}) + \boldsymbol{\epsilon}$, where $f$ is a deterministic function that combines latent confounders $\boldsymbol{C}$ and latent post-treatment variables $\boldsymbol{M}$ into observations $\boldsymbol{X}$ and $\boldsymbol{\epsilon}$ is random noise. In addition, we assume that the function $f$ is **injective**; beyond injectivity, $f$ can be arbitrarily nonlinear. We use $f^\dagger : \boldsymbol{X} \to [\boldsymbol{C}||\boldsymbol{M}]$ to denote its left inverse. We use $f_C^\dagger : \boldsymbol{X} \to \boldsymbol{C}$ and $f_M^\dagger : \boldsymbol{X} \to \boldsymbol{M}$ to denote the mapping from $\boldsymbol{X}$ to $\boldsymbol{C}$, $\boldsymbol{M}$, respectively.*

***Noisy-Injectivity*** is a common assumption made either explicitly or implicitly in most existing proxy-of-confounder-based causal inference algorithms. For example, if both $\boldsymbol{X}$ and $\boldsymbol{C}$ are categorical, Pearl (2012) assumes that $\boldsymbol{X}$ has at least the same number of categories as $\boldsymbol{C}$, whereas the effect restoration algorithm (Rothman et al., 2008) assumes that the matrix of $p(\boldsymbol{C}, \boldsymbol{X})$ to be full-rank. Although CEVAE (Louizos et al., 2017) makes no explicit injectivity assumption between $\boldsymbol{C}$ and $\boldsymbol{X}$, it requires that the joint distribution $p(\boldsymbol{C}, \boldsymbol{X}, T, Y)$ can be fully recovered from the observations $(\boldsymbol{X}, T, Y)$. The literature shows that some of the possible identification criteria are **1)** multiple independent views of $\boldsymbol{C}$ in $\boldsymbol{X}$ (Edwards et al., 2015), and **2)** $\boldsymbol{C}$ is categorical and $\boldsymbol{X}$ is a mixture of Gaussian components determined by $\boldsymbol{C}$ (that is, $\boldsymbol{X}$ is generated by bijective mapping of $\boldsymbol{C}$ to the mean of the corresponding component with added Gaussian noise) (Anandkumar et al., 2014).

In the following part of this section, we omit the noise $\boldsymbol{\epsilon}$ to gain better intuition of latent post-treatment bias (but all the exact conclusions will still hold in the posterior sense). In Section 3, we assume that noise exists and demonstrate that our method can still adequately identify latent confounders.

#### 2.2.2 CAUSAL ESTIMAND AND THE TRUE ATE

Based on Assumption 1, we are ready to define the estimated average treatment effect (ATE) by controlling the covariates $\boldsymbol{X}$, as well as the true (conditional) average treatment effects.

**Definition 1.** *We define the Difference in Conditional Expected Values (DCEV) as*

$$DCEV(\boldsymbol{x}) = \mathbb{E}[Y|T = 1, \boldsymbol{X} = \boldsymbol{x}] - \mathbb{E}[Y|T = 0, \boldsymbol{X} = \boldsymbol{x}], \qquad (1)$$

*which is the difference of the expected value of the outcome $Y$ for units with variable $\boldsymbol{X} = \boldsymbol{x}$ in the treatment group and the non-treatment group. Based on $DCEV(\boldsymbol{x})$, we define the Difference in Expected Value (DEV), i.e., $DEV(\boldsymbol{X}) = \mathbb{E}_{p(\boldsymbol{X})}[DCVE(\boldsymbol{X})]$ as the expected value DCEV.*

$DEV(\boldsymbol{X})$ denotes the ATE estimand by controlling covariates $\boldsymbol{X}$. If $\boldsymbol{X} = \emptyset$, $DEV(\emptyset)$ represents the *naive estimator* that directly calculates the expected difference of $Y$ between the treatment group and the non-treatment group. With the causal estimand $DEV(\boldsymbol{X})$ introduced, we then define the true causal effects (i.e., C/ATE) when covariates $\boldsymbol{X}$ scramble both latent confounders and post-treatment variables according to the generative process described in Assumption 1. The main issue that hinders a direct definition of C/ATE with $DCEV(\boldsymbol{x})$ and $DEV(\boldsymbol{X})$ is that, since $\boldsymbol{X}$ contains latent post-treatment variables $\boldsymbol{M}$, conditional on $\boldsymbol{X}$, the strong ignorability assumption (Imbens & Rubin, 2015) widely used for the identification of causal effects **does not hold**[2]. Accordingly, we have:

**Definition 2.** *Under Assumption 1, we define the Conditional Average Treatment Effect (CATE) for individuals with observed covariates $\boldsymbol{X} = \boldsymbol{x}$ as follows:*

$$CATE(\boldsymbol{x}) = \mathbb{E}[Y|T = 1, \boldsymbol{C} = f_C^\dagger(\boldsymbol{x})] - \mathbb{E}[Y|T = 0, \boldsymbol{C} = f_C^\dagger(\boldsymbol{x})], \qquad (2)$$

*with the Average Treatment Effect (ATE) of treatment $T$ defined as*

$$ATE = \mathbb{E}[Y|do(T = 1)] - \mathbb{E}[Y|do(T = 0)] = \mathbb{E}_{p(\boldsymbol{C})}[\mathbb{E}[Y|T = 1, \boldsymbol{C}] - \mathbb{E}[Y|T = 0, \boldsymbol{C}]]. \quad (3)$$

In Definition 2, we only consider the latent confounder component of $\boldsymbol{X}$ for CATE in Eq. (2), as the causal relationship between the post-treatment variables $\boldsymbol{M}$ and the outcome $Y$ is indeterminate (see Fig. 1-(c)). However, if the specific relationship between $\boldsymbol{M}$ and $Y$ can be further established by the researcher (e.g., all elements of $\boldsymbol{M}$ are latent mediators), more precise forms of CATE can be derived with path-specific counterfactual analysis (Imai et al., 2010; Cheng et al., 2022).

### 2.2.3 LATENT POST-TREATMENT BIAS

With $DEV(\boldsymbol{X})$ (the ATE estimator that controls the covariates $\boldsymbol{X}$), CATE, and ATE defined in Section 2.2.2, in this section, we analyze the *latent post-treatment bias* of existing proxy-of-confounder-based causal inference methods, such as CEVAE (Louizos et al., 2017), that control latent variables inferred from the covariates $\boldsymbol{X}$ to estimate the ATE of $T$ on $Y$, when $\boldsymbol{X}$ scrambles both latent confounders and post-treatment variables. In our analysis, Lemma 2.1 will be frequently used.

**Lemma 2.1.** *For an injective function $g$, $\mathbb{E}[Y|\boldsymbol{X} = \boldsymbol{x}] = \mathbb{E}[Y|g(\boldsymbol{X}) = g(\boldsymbol{x})]$ holds.*

The proof when $g$ is differentiable *a.e.* can be referred to in Appendix A.1. Since the latent variable models used in existing methods (such as VAE with factorized Gaussian prior in CEVAE) lack identifiability guarantee (i.e., the recovery of the exact latent variables), we assume that these models can recover the true latent space $\boldsymbol{Z} = [\boldsymbol{C}, \boldsymbol{M}]$ up to invertible transformations $\bar{f}$, where the inference process can be represented as $\hat{\boldsymbol{Z}} = \tilde{f}(\boldsymbol{X}) = \bar{f} \circ f^\dagger(\boldsymbol{X})$. With such an assumption, we have the following theorem regarding the latent post-treatment bias when $\boldsymbol{X}$ mixes post-treatment variables.

**Theorem 2.2.** *If the observed covariates $\boldsymbol{X}$ are generated from latent confounders $\boldsymbol{C}$ and latent post-treatment variables $\boldsymbol{M}$ according to Assumption 1, the latent post-treatment bias of a proxy-of-confounder-based causal inference algorithm that controls latent variables $\hat{\boldsymbol{Z}}$ inferred from $\boldsymbol{X}$ via $\tilde{f} = \bar{f} \circ f^\dagger : \mathbb{R}^{K_X} \to \mathbb{R}^{K_C + K_M}$ to estimate the ATE can be formulated as follows:*

$$\begin{aligned} Bias(\boldsymbol{X}) &= ATE - DEV(\tilde{f}(\boldsymbol{X})) = ATE - \mathbb{E}[\mathbb{E}[Y|T = 1, \tilde{f}(\boldsymbol{X})] - \mathbb{E}[Y|T = 0, \tilde{f}(\boldsymbol{X})]] \\ &= ATE - \mathbb{E}[\mathbb{E}[Y|T = 1, \bar{f} \circ f^\dagger(f(\boldsymbol{C}, \boldsymbol{M}))] - \mathbb{E}[Y|T = 0, \bar{f} \circ f^\dagger(f(\boldsymbol{C}, \boldsymbol{M}))]] \\ &= \mathbb{E}[\mathbb{E}[Y|T = 1, \boldsymbol{C}] - \mathbb{E}[Y|T = 0, \boldsymbol{C}]] - \mathbb{E}[\mathbb{E}[Y|T = 1, \boldsymbol{C}, \boldsymbol{M}] - \mathbb{E}[Y|T = 0, \boldsymbol{C}, \boldsymbol{M}]], \end{aligned}$$
$$(4)$$

*which can be arbitrarily bad. Therefore, the estimator of existing proxy-of-confounder-based methods, i.e., $DEV(\tilde{f}(\boldsymbol{X}))$, is an arbitrarily biased estimator of the ATE, when the selected proxy of confounders $\boldsymbol{X}$ accidentally mixes in latent post-treatment variables $\boldsymbol{M}$.*

---

[2]Equivalently, we could say that given covariates $\boldsymbol{X}$, the **backdoor criteria** between $T$ and $Y$ does not hold, which requires the conditional set of variables contains no descendants of the treatment $T$ (Glymour et al., 2016).

The final step of Eq. (4) can be proved since $f$ is injective and $\bar{f}$ bijective, the composite $\bar{f} \circ f^{\dagger} \circ f : [\boldsymbol{C}, \boldsymbol{M}] \to \hat{\boldsymbol{Z}}$ is bijective, so we can use Lemma 2.1 to remove $\bar{f} \circ f^{\dagger} \circ f$ in the condition.

### 2.2.4 EXAMPLES IN THE LINEAR CASES

Generally, the latent post-treatment bias defined in Eq. (4) cannot be simplified because **1)** the causal relationship between $\boldsymbol{M}$ and $Y$ is indeterminate, and **2)** the causal influence of $\boldsymbol{C}$, $\boldsymbol{M}$, and $T$ on $Y$ can be arbitrary. However, for linear structural causal models with causal relationships determined between $\boldsymbol{M}$ and $Y$ (e.g., $\boldsymbol{M}$ are mediators, which are post-treatment variables that have causal influences on the outcomes), stronger conclusions can be drawn as follows:

**Corollary 2.3.** *(MixedMediator) For the linear Structural Causal Model (SCM) defined as:*

$$
\begin{aligned}
T &\leftarrow \mathbb{1}(\alpha_T + \sum \beta_i \cdot C_i > a) \\
M_j &\leftarrow \alpha_M + \gamma_j \cdot T \\
\boldsymbol{X} &\leftarrow \boldsymbol{\alpha}_X + \mathbf{A}[\boldsymbol{M} \| \boldsymbol{C}] \\
Y &\leftarrow \alpha_Y + \tau \cdot T + \sum \theta_j \cdot M_j + \sum \kappa_i \cdot C_i,
\end{aligned}
\tag{5}
$$

*where the mixture function $f = \mathbf{A} \in \mathbb{R}^{K_X \times (K_C + K_M)}$ is a full column-rank matrix, the CATE, ATE, and the bias of proxy-of-confounder-based causal inference model that controls the latent variables $\hat{\boldsymbol{Z}}$ inferred via $\hat{\boldsymbol{Z}} = \tilde{f}(\boldsymbol{X}) = \mathbf{B}^T \boldsymbol{X}$ can be formulated as follows:*

$$
\begin{aligned}
ATE &= CATE = \tau + \sum \gamma_j \cdot \theta_j \\
DEV(\hat{\boldsymbol{Z}}) &= \mathbb{E}[DCEV(\hat{\boldsymbol{Z}})] = DCEV(\hat{\boldsymbol{Z}}) = \tau \\
Bias(\hat{\boldsymbol{Z}}) &= ATE - DEV(\hat{\boldsymbol{Z}}) = \sum \gamma_j \cdot \theta_j,
\end{aligned}
\tag{6}
$$

*where $\mathbf{B} \in \mathbb{R}^{K_X \times (K_C + K_M)}$ is another full column-rank matrix. Since $\sum \gamma_j \cdot \theta_j$ is arbitrary, the estimator $DEV(\hat{\boldsymbol{Z}}) = \mathbb{E}[DCEV(\mathbf{B}^T \boldsymbol{X})]$ is arbitrarily biased for ATE estimation.*

The proof of Eq. (6) is provided in Appendix A.2. In addition, we show that the post-treatment variables $\boldsymbol{M}$ DO NOT necessarily need to have direct causal effects on the outcome $Y$ to incur arbitrary bias in ATE estimation. In Appendix A.3, we provide another example (i.e., MixedCorrelator) in the linear case where $\boldsymbol{M}$ is correlated with $Y$ through unobserved confounders $\boldsymbol{U}$ in Corollary A.1.

## 3 METHODOLOGY

In this section, we introduce the proposed Confounder-identifiable Variational Auto-Encoder (**CiVAE**) to address latent post-treatment bias. Specifically, we first prove that if the prior distribution of the true latent variables $\boldsymbol{Z} = [\boldsymbol{C}, \boldsymbol{M}]$ satisfies certain weak assumptions, identifiability criterion holds, and each dimension of the inferred latent variables $\hat{\boldsymbol{Z}}$, i.e., $\hat{Z}_i$, corresponds to the invertible transformation of **either** a true confounder $C_j$ **or** a true post-treatment variable $M_k$. Then, utilizing the causal relations between $\boldsymbol{C}$, $\boldsymbol{M}$, and $T$, we novelly transform the challenging confounder-identifiability problem into a tractable pair-wise conditional independence test problem, which can be effectively solved with kernel-based methods. Finally, we demonstrate that controlling the transformed confounders inferred by CiVAE can yield an unbiased estimation of the true ATE.

### 3.1 GENERATIVE PROCESS

The fundamental work of deep variational inference with identifiability guarantee, i.e., the identifiable VAE (iVAE) (Khemakhem et al., 2020), makes a strict assumption that the prior of true latent variables $\mathbf{Z}$ (i.e., $[\boldsymbol{C}, \boldsymbol{M}]$ in our case) is conditionally factorized given the available covariates (i.e., the treatment $T$ and the outcome $Y$ in our case). However, since both latent confounders $\boldsymbol{C}$ and latent post-treatment variables $\boldsymbol{M}$ form fork structures with the outcome $Y$ (see Fig. 1-(c)) (Koller & Friedman, 2009), $C_i$, $C_j$, $M_i$, and $M_j$ are not independent given $Y$. Recently, Non-Factorized iVAE (NF-iVAE) (Lu et al., 2021) was proposed that allows arbitrary dependence among the true latent variables $\boldsymbol{Z}$ in the conditional priors, where $\boldsymbol{Z}$ can be identified up to arbitrary non-linear

transformations, However, the transformation are not necessarily invertible, which is risky for causal inference, as multiple values of the confounders may collapse, leading to bias when estimating the ATE by averaging the $DCEV$ calculated in each stratum of the inferred confounders.

The proposed NF-iVAE guarantees the identifiability of $\mathbf{Z}$ by putting a general exponential family distribution with at least one invertible sufficient statistic in the factorized part as its prior when conditioning on treatment $T$ and outcome $Y$, which can be formulated as follows.

**Assumption 2.** *Let* $\mathbf{Z} = [\mathbf{C}\|\mathbf{M}]$ *be the random vector for latent variables that causally generate the observed covariates* $\mathbf{X}$ *according to Assumption 1. We assume that the conditional prior of* $\mathbf{Z}$ *given the outcome* $Y$ *and the treatment* $T$ *belongs to a general exponential family with parameter vector* $\boldsymbol{\lambda}(Y,T)$ *and sufficient statistics* $\mathbf{S}(\mathbf{Z}) = [\mathbf{S}_f(\mathbf{Z})^T, \mathbf{S}_{nf}(\mathbf{Z})^T]^T$. *Specifically,* $\mathbf{S}(\mathbf{Z})$ *is composed of (i) the sufficient statistics of a factorized exponential family, i.e.,* $\mathbf{S}_f(\mathbf{Z}) = [\mathbf{S}_1(Z_1)^T, \cdots, \mathbf{S}_{K_Z}(Z_{K_Z})^T]^T$, *where all components* $\mathbf{S}_i(Z_i)$ *have dimension larger than or equal to 2 and **each** $\mathbf{S}_i$ **has at least one invertible dimension**, and (ii)* $\mathbf{S}_{nf}(\mathbf{Z})$, *where* $\mathbf{S}_{nf}$ *is a neural network with ReLU activation. The density of the conditional prior can be formulated as:*

$$p_{\mathbf{S},\boldsymbol{\lambda}}(\mathbf{Z}|Y,T) = \mathcal{Q}(\mathbf{Z})/\mathcal{C}(Y,T)\exp[\mathbf{S}(\mathbf{Z})^T\boldsymbol{\lambda}(Y,T)], \tag{7}$$

*where* $\mathcal{Q}(\mathbf{Z})$ *is the base measure and* $\mathcal{C}(Y,T)$ *not dependent on* $\mathbf{Z}$ *is the normalizing constant.*

We justify that assumption 2 is weak and practical as follows. **1)** Neural networks with ReLU activation have universal approximation ability of distributions (Lu & Lu, 2020). Therefore, Eq. (7) can model arbitrary dependence between true latent confounders $\mathbf{C}$ and true post-treatment variables $\mathbf{M}$ conditional on $T$ and $Y$. **2)** Although CiVAE makes an extra assumption that $\forall i$, at least one dimension of $\mathbf{S}_i$ is invertible, this can be easily satisfied as most commonly used exponential family distributions, such as Gaussian, Bernoulli, etc., has at least one invertible sufficient statistics[3].

The reason why we use ReLU as the activation is that, the identifiability of iVAE relies on the condition that the sufficient statistics $\mathbf{S}$ have zero second-order cross-derivative. The factorized part, i.e., $\mathbf{S}_f$, satisfies it trivially since all cross-derivatives of $\mathbf{S}_f$ are zero. In addition, since the ReLU neural networks are linear *a.e.*, all second-order derivatives of $\mathbf{S}_{nf}$ are zero. Therefore, identifiability holds after adding $\mathbf{S}_{nf}$ in the prior that allows the capturing of arbitrary dependence among $\mathbf{Z}$.

## 3.2 OPTIMIZATION OBJECTIVE

Combining Assumptions 1 and 2, the generative process of CiVAE can be formulated as follows:

$$p_{\boldsymbol{\theta}}(\mathbf{X}, \mathbf{Z} \mid Y, T) = p_f(\mathbf{X} \mid \mathbf{Z})p_{\mathbf{S},\boldsymbol{\lambda}}(\mathbf{Z} \mid Y, T), \tag{8}$$

$$p_f(\mathbf{X} \mid \mathbf{Z}) = p_{\boldsymbol{\epsilon}}(\mathbf{X} - f(\mathbf{Z})). \tag{9}$$

where $\boldsymbol{\theta} = (f, \boldsymbol{\lambda}, \mathbf{S}) \in \Theta$ are the parameters of the generative distribution[4]. Since the generative process of CiVAE is parameterized by deep neural networks, the posterior distribution of $\mathbf{Z}$, i.e., $p_{\boldsymbol{\theta}}(\mathbf{Z} \mid \mathbf{X}, Y, T)$, is intractable. Therefore, we resort to variational inference (Blei et al., 2017), where we introduce approximate posterior $q_{\boldsymbol{\phi}}(\mathbf{Z} \mid \mathbf{X}, Y, T)$ parameterized by deep neural network with trainable parameter $\boldsymbol{\phi}$, and in $q_{\boldsymbol{\phi}}(\mathbf{Z}|\cdot)$ finds the one closes to $p_{\boldsymbol{\theta}}(\mathbf{Z}|\cdot)$ measured by KL divergence. Minimization of the KL is equivalent to maximization of the evidence lower bound (ELBO) as:

$$\mathcal{L}(\boldsymbol{\theta}, \boldsymbol{\phi}) := \mathbb{E}_{q_{\boldsymbol{\phi}}(\mathbf{Z}|\mathbf{X},Y,T)}\left[\log p_f(\mathbf{X} \mid \mathbf{Z}) + \underbrace{\log p_{\mathbf{S},\boldsymbol{\lambda}}(\mathbf{Z} \mid Y, T) - \log q_{\boldsymbol{\phi}}(\mathbf{Z} \mid \mathbf{X}, Y, T)}_{\text{KL of posterior with prior}}\right]. \tag{10}$$

Since the normalization constant $\mathcal{C}$ in Eq. (7) is generally intractable, it is infeasible to directly learn $\mathbf{S}, \boldsymbol{\lambda}$ by optimizing Eq. (10). Therefore, we substitute the KL term in Eq. (10) with the widely-used score matching (Hyvärinen & Dayan, 2005) to learn unnormalized densities instead as follows:

$$\mathcal{L}(\mathbf{S}, \boldsymbol{\lambda}, \boldsymbol{\phi}) := \mathbb{E}_{q_{\boldsymbol{\phi}}(\mathbf{Z}|\mathbf{X},Y,T)}\left[\|\nabla_{\mathbf{Z}}\log q_{\boldsymbol{\phi}}(\mathbf{Z} \mid \mathbf{X}, Y, T) - \nabla_{\mathbf{Z}}\log p_{\mathbf{S},\boldsymbol{\lambda}}(\mathbf{Z} \mid Y, T)\|^2\right]$$

$$= \mathbb{E}_{q_{\boldsymbol{\phi}}(\mathbf{Z}|\mathbf{X},Y,T)}\left[\sum_{j=1}^{K_Z}\left[\frac{\partial^2 p_{\mathbf{S},\boldsymbol{\lambda}}(\mathbf{Z} \mid Y, T)}{\partial Z_j^2} + \frac{1}{2}\left(\frac{\partial p_{\mathbf{S},\boldsymbol{\lambda}}(\mathbf{Z} \mid Y, T)}{\partial Z_j}\right)^2\right]\right] + \text{const.} \tag{11}$$

---

[3]There are a few exponential family with no invertible sufficient statistics, e.g., Weibull distribution when shape parameter $k$ is even.

[4]Note that although $f$ is a function, we include it in the parameter set to be consistent with the iVAE paper.

### 3.3 IDENTIFIABILITY OF CIVAE

With the generative process and optimization objective of CiVAE introduced in the previous sub-sections, we are ready to introduce the final assumption of CiVAE, which, combined with Assumptions 1 and 2, leads to the main theorem of this paper, which states the identifiability of CiVAE.

**Assumption 3.** *Assume the following: (i) The set $\{\boldsymbol{X} \in \mathcal{X} | \phi(\boldsymbol{X}) = 0\}$ has measure zero, where $\phi$ is the characteristic function of the density $p_f$ in Eq. (9). (ii) The sufficient statistics, $\boldsymbol{S}_i$ in $\boldsymbol{S}_f$ are all twice differentiable. (iii) The mixture function $f$ in Eq. (9) has all second-order cross derivatives. (iv) There exist $k+1$ distinct points $(Y, T)_0, \cdots, (Y, T)_k$ such that the matrix $\mathbf{L} = [\boldsymbol{\lambda}((Y, T)_1) - \boldsymbol{\lambda}((Y, T)_0), \cdots, \boldsymbol{\lambda}((Y, T)_k) - \boldsymbol{\lambda}((Y, T)_0)]$ of size $k \times k$ is invertible, where $k = Dim(\boldsymbol{S})$.*

*(i) - (iii)* are trivial for neural networks. *(iv)* denotes that independent samples of $(Y, T)$ are required to identify $\boldsymbol{C}$ and $\boldsymbol{M}$. The identifiability theorem of CiVAE can be formulated as follows.

**Theorem 3.1.** *If Assumptions 1, 2, and 3 hold, and if $\boldsymbol{\theta}, \tilde{\boldsymbol{\theta}} \in \Theta \to p_{\boldsymbol{\theta}}(\boldsymbol{X}|Y, T) = p_{\tilde{\boldsymbol{\theta}}}(\boldsymbol{X}|Y, T)$, the true latent variables $\boldsymbol{Z}$ are identifiable up to **permutation** and **element-wise bijective transformation**. Furthermore, in the case of **variational inference**, if we denote the true parameter that generates the data as $\boldsymbol{\theta}^*$, if (i) the distribution family $q_{\boldsymbol{\phi}}(\boldsymbol{Z}|\boldsymbol{X}, Y, T)$ contains the posterior $p_{\boldsymbol{\theta}}(\boldsymbol{Z}|\boldsymbol{X}, Y, T)$, and $q_{\boldsymbol{\phi}}(\boldsymbol{Z}|\boldsymbol{X}, Y, T) > 0$, (ii) we optimize Eq. (4) w.r.t. both $\boldsymbol{\theta}, \boldsymbol{\phi}$, then in the limit of infinite data, true parameters $\boldsymbol{\theta}^*$ can be learned up to a permutation and bijective transformation of $\boldsymbol{Z}$.*

The proof of Theorem 3.1 is based on the NF-iVAE paper (Lu et al., 2021), with the new assumption introduced in CiVAE that each $\boldsymbol{S}_i$ has at least one invertible dimension incorporated to ensure that the transformation of each $Z_i$ is bijective. The detailed proof is provided in Appendix A.4.

### 3.4 IDENTIFICATION OF LATENT CONFOUNDERS

Theorem 3.1 ensures that latent variables $\hat{\boldsymbol{Z}}$ inferred by CiVAE cannot **1)** mix confounders and post-treatment variables in each dimension, or **2)** collapse different values of the latent confounders into the same value. To further determine the dimensions of confounder and post-treatment variable in $\hat{\boldsymbol{Z}}$, we rely on the causal relations between latent variables $\boldsymbol{Z} = [\boldsymbol{C}, \boldsymbol{M}]$ and treatment $T$ and the associated marginal/conditional dependence properties. These are discussed as follows.

- *Case 1. Intra-Confounders.* Latent confounders $C_i$, $C_j$ and the treatment $T$ form the *V-structure* $C_i \to T \leftarrow C_j$. Therefore, $C_i$ and $C_j$ are marginally **independent**, whereas they become **dependent** when conditioning on the assigned treatment $T$.
- *Case 2. Intra-Post Treatment Variables.* Latent post-treatment variables $M_i$, $M_j$ and the treatment $T$ form a *fork-structure* $M_i \leftarrow T \to M_j$, where $M_i$, $M_j$ are marginally **dependent**, but they become **independent** after conditioning on the assigned treatment $T$.
- *Case 3. Cross-Confounder and Post-Treatment Variables.* Latent confounder $C_i$, latent post-treatment variable $M_j$, and the treatment $T$ forms a *chain structure* $C_i \to T \to M_j$, where $C_i$, $M_j$ are marginally dependent, and they become **independent** after conditioning on $T$.

From the above analysis we can find that, the dependence between two latent variables $Z_i$ and $Z_j$ **increases** after conditioning on the treatment $T$ ONLY in the case of *intra-confounders*. Therefore, if more than one latent confounders exist, which is highly probable when covariates $\boldsymbol{X}$ are high-dimensional, we can conduct independence test $\texttt{Ind}(\hat{Z}_i, \hat{Z}_j)$ and $\texttt{CInd}(\hat{Z}_i, \hat{Z}_j|T)$ for all pairs of inferred latent variables, which can be implemented via kernel-based methods as (Zhang et al., 2012), and select the pairs where p-value of $\texttt{CInd}$ is larger than that of $\texttt{Ind}$ as latent confounders.

Here, we note that the kernel-based (conditional) independence test incurs $N^2 \times K_Z^2$ complexity in the training phase. However, once the dimensions of the confounders in $\hat{\boldsymbol{Z}}$ are determined, CiVAE **has the same complexity as CEVAE** for the estimation of CATE and ATE in the test phase. Therefore, we argue that the additional complexity of model training is worthy due to the substantially increased robustness toward latent post-treatment bias (which will be demonstrated in Section 4).

### 3.5 ATE ESTIMATOR WITH TRANSFORMED CONFOUNDERS

Finally, we show that controlling transformed confounders $\hat{\boldsymbol{C}}$ inferred by CiVAE provides an unbiased estimation of ATE. Although assumptions weaker than Assumption 2, e.g., inferred confounders have

the same propensity score as the true confounders (i.e., $\hat{C}$ does not have to be bijective transformation of $C$), could lead to the same unbiasedness results (Imbens & Rubin, 2015), since our main purpose is to analyze the latent post-treatment bias and propose a viable solution accordingly, this introduces unnecessary complexity, which could be explored as a direction for future study.

**Theorem 3.2.** *Controlling bijective of confounders is equivalent to controlling true confounders in ATE estimation, i.e., $DEV(\hat{C}) = DEV(g(C)) = ATE$, if transformation function $g$ is bijective.*

The proof of Theorem 3.2 for discrete $C$ is trivial (where $\hat{C} = g(C)$ represents a simple relabeling of the stratum that we calculate the $DCEV$ and take the expectation). The proof in the continuous case where $g$ is differentiable is provided in Appendix A.5. With Theorem 3.2, we can control the identified latent confounders as true confounders, providing an unbiased estimate of ATE.

## 4 EMPIRICAL STUDY

### 4.1 DATASETS

We establish two simulated datasets, i.e., `MixedMediator` and `MixedCorrelator`, that consider two types of post-treatment variables, i.e., **1)** mediators and **2)** variables that are correlated with the outcome $Y$ via latent confounders $U$. The generative process of the two datasets can be referred to in Corollary 2.3 and Corollary A.1, respectively, where the latent confounders $C$ are generated from Gaussian as $C \sim Gaussian(0, \mathbf{I}_{K_C})$. For `MixedMediator`, $\gamma$ is set as $[-1, -1, -1]$, $\theta$ is set as $[1, 1, 1]$, and $\tau$ is set as 2, which results in $ATE = -1$. For `MixedMediator`, we set the same $\gamma$ and $\theta$ as `MixedMediator`, where parameters $\phi = 1$ and $\tau = 1$, which results in $ATE = 1$.

In addition, we build a real-world dataset based on the job Ads data from the Company, aiming to estimate the ATE of *switching a job from **onsite** to **online** working mode* to *the statistics of the applicants* (here we choose the average age as the outcome). In the dataset, treatment $T$ represents the working mode of the job, where $T = 1$ represents the job is online, whereas $T = 0$ represents the job is onsite, $Y$ is the standardized age, and $X \in \{0, 1\}^{K_X}$ indicates the required skills of the job. We select 3,228 jobs from Bay Area, where a primary study shows that $DEV(\emptyset) \approx -2$ years[5] (i.e., online job applicants are two years younger than onsite job applicants). To simulate the latent confounder $C$ and post-treatment variables $M$, we first learn a generative model as follows:

$$Z \sim Gaussian(\mathbf{0}, \mathbf{I}_{K_Z}), X \sim Multi(NN_f(Z)), Y \sim Gaussian(w \odot Z, 1) \qquad (12)$$

where $Multi$ represents multinomial distribution, $NN_f$ is a neural network with softmax activation, $Z, w \in \mathbb{R}^{K_Z}$, $K_Z = 6$, and $\odot$ represents the element-wise product operator, respectively. We then treat the first $K_C = 3$ dimensions of $Z$ as the latent confounders $C$ and the remaining $K_M = K_Z - K_C$ dimensions as the latent mediators $M$. After learning $NN_f$ and $w$ according to Eq. (12), we draw latent confounders $C \in Gaussian(0, \mathbf{I})$, latent mediators $M = T \cdot \gamma$, and set the outcome $Y = w \odot [C || M] + \tau \cdot T$, where the true ATE can be calculated as $sum(\gamma \odot w_{-K_M:}) + \tau$.

### 4.2 COMPARISONS WITH THE STATE-OF-THE-ART

The baselines we include for comparisons can be categorized into three classes. **1) Unawareness**, where no information in $X$ is used for ATE estimation. We implement the naive LR0 estimator, which regresses $Y$ on $T$ and uses the coefficient to estimate the ATE (Imbens & Rubin, 2015) (LR0 equals to $DEV(\emptyset)$, i.e., the difference of average outcome between the treatment and non-treatment group). **2) Control-$X$**, which directly controls the covariates $X$. In this class, LR1 regresses $Y$ on $T$ and $X$, whereas TarNet uses a two-branch neural network to estimate the $DEV(X)$ **3) Control-$Z$**, which controls latent variables $Z$ learned from the covariates $X$. Methods from this class include the CEVAE (Louizos et al., 2017) and covariate disentanglement methods (see Fig. 1-(b)), such as DR-CFR Hassanpour & Greiner (2020) and TEDVAE (Zhang et al., 2021).

The comparisons are summarized in Table 1. From Table 1, we can empirically verify the correctness of Theorem 2.2 that post-treatment bias indeed poses a serious issue for proxy-of-confounder-based methods, because for the `MixedMediator` and `MixedConfounder` datasets, CEVAE is worse than the naive LR0 estimator that directly calculates the difference of mean outcome between the

---

[5]which leads to -0.178 after standardization. Code demo see `https://anonymous.4open.science/r/CiVAE-demo-54B9`.

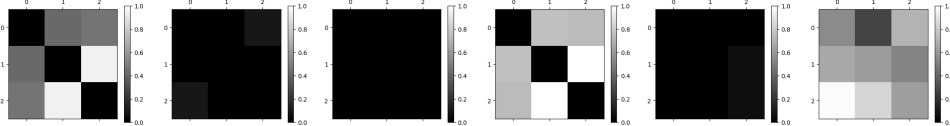

(a) **Case 1:** Intra-Confounder      (b) **Case 2:** Intra-Mediator      (c) **Case 3:** Confounder-Mediator

Figure 2: Visualization of $p$-value of independence test before and after conditioning on treatment $T$.

Table 1: Comparison of CiVAE with baselines on ATE estimation with latent post-treatment bias.

| Dataset | MixedMediator | | MixedCorrelator | | Company | |
|---|---|---|---|---|---|---|
| Method | ATE. | Err. | ATE. | Err. | ATE. | Err. |
| LR0 | $0.975 \pm_{0.032}$ | 1.975 | $2.977 \pm_{0.032}$ | 1.977 | $0.131 \pm_{0.015}$ | 0.399 |
| LR1 | $1.457 \pm_{0.167}$ | 2.457 | $3.400 \pm_{0.130}$ | 2.400 | $0.093 \pm_{0.071}$ | 0.361 |
| TarNet | $1.461 \pm_{0.172}$ | 2.461 | $3.414 \pm_{0.146}$ | 2.414 | $0.112 \pm_{0.085}$ | 0.380 |
| CEVAE | $1.550 \pm_{0.292}$ | 2.550 | $3.323 \pm_{0.167}$ | 2.323 | $0.106 \pm_{0.078}$ | 0.374 |
| DR-CFR | $1.239 \pm_{0.324}$ | 2.239 | $3.185 \pm_{0.319}$ | 2.185 | $0.094 \pm_{0.089}$ | 0.362 |
| TEDVAE | $1.042 \pm_{0.315}$ | 2.042 | $3.138 \pm_{0.281}$ | 2.138 | $0.097 \pm_{0.093}$ | 0.365 |
| CiVAE | $\mathbf{-0.822} \pm_{0.753}$ | **0.178** | $\mathbf{1.199} \pm_{0.765}$ | **0.199** | $\mathbf{-0.140} \pm_{0.137}$ | **-0.128** |
| True ATE | $-1.000 \pm_{0.000}$ | 0.000 | $1.000 \pm_{0.000}$ | 0.000 | $-0.268 \pm_{0.000}$ | 0.000 |

treatment and non-treatment groups. In addition, for `MixedMediator` and `Company` datasets, all methods except the proposed CiVAE fail to predict the negativity of the ATE.

Covariates disentanglement-based methods, i.e., DR-CFR and TEDVAE, achieve similar performance as CEVAE. The reason is that, these methods disentangle latent confounders $C$ from latent instrumental variables $I$ and latent adjusters $A$ by utilizing their causal relations with $T$ and $Y$, i.e., $I$ is predictive only for $T$, $A$ is predictive only for $Y$, whereas $C$ is predictive for both $T$ and $Y$. For example, TEDVAE includes three encoders to infer three sets of latent variables $\hat{I}, \hat{A}, \hat{C}$ from $X$ and adds classification losses $p(T|\hat{I}, \hat{C})$ and $p(Y|T, \hat{C}, \hat{A})$ on the CEVAE loss. However, when latent post-treatment bias exists, since both latent confounders $C$ and latent post-treatment variables $M$ are correlated with both $T$ and $Y$, $\hat{C}$ inferred by TEDVAE still cannot disentangle $C$ from $M$.

CiVAE achieves significantly better results compared to CEVAE and TEDVAE, which demonstrates its effectiveness in identifying and distinguishing latent confounders from post-treatment variables in proxies. However, we also notice that a downside of CiVAE is the comparatively large variance across ten dataset splits, as misidentifying latent mediators as confounders may result in severe performance degradation when the mediation effects are strong or the number of latent confounders is small.

### 4.3   DISENTANGLING OF LATENT CONFOUNDERS AND POST-TREATMENT VARIABLES

We show the $p$-value of the pairwise independence test of the true latent variables before and after conditioning on the assigned treatment $T$. From Fig. 2 we can find that the difference between the three cases discussed in Subsection 3.4 is significant. Here, we should note that the distinction of the intra-confounder case from other cases relies on the assumption that latent confounders are independent. If the latent confounders are correlated, we can first use causal discovery techniques such as the PC algorithm (Spirtes et al., 2000) to find direct parents of $T$, and use our algorithm as the refinement to determine the true confounders $C$ from the misidentified post-treatment variables.

## 5   CONCLUSIONS

In this paper, we systematically investigated the latent post-treatment bias in causal inference from observational data. We first prove that unresolved latent post-treatment variables scrambled in the proxy of confounders can arbitrarily bias the ATE estimation. To address the bias, we proposed the Confounder-identifiable VAE (CiVAE), which, utilizing a mild assumption regarding the prior of latent factors, guarantees the identifiability of latent confounders up to bijective transformations. Finally, we show that controlling the latent confounders inferred by CiVAE can provide an unbiased estimation of the ATE. Experiments on both simulated and real-world datasets demonstrated that CiVAE has superior robustness to latent post-treatment bias compared with state-of-the-art methods.

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

# A PROOFS

## A.1 PROOF OF LEMMA 2.1.

*Proof.* Let $\boldsymbol{Z} = g(\boldsymbol{X})$ and $\boldsymbol{z} = g(\boldsymbol{x})$. If $g$ is injective and differentiable *a.e.*, and $g^\dagger$ is the left-inverse, we have:

$$f_{Y|g(\boldsymbol{X})}(y|g(\boldsymbol{x})) = f_{Y|\boldsymbol{Z}}(y|\boldsymbol{z}) = \frac{f_{Y,\boldsymbol{Z}}(y,\boldsymbol{z})}{f_{\boldsymbol{Z}}(\boldsymbol{z})} = \frac{f_{Y,\boldsymbol{X}}(y,g^\dagger(\boldsymbol{z}))|\mathbf{J}_{g^\dagger}(\boldsymbol{z})|}{f_{\boldsymbol{X}}(g^\dagger(\boldsymbol{z}))|\mathbf{J}_{g^\dagger}(\boldsymbol{z})|} = \frac{f_{Y,\boldsymbol{X}}(y,\boldsymbol{x})}{f_{\boldsymbol{X}}(\boldsymbol{x})} = f_{Y|\boldsymbol{X}}(y|\boldsymbol{x}),$$
$$(13)$$

where $f_.$ and $f_{.|.}$ represent the marginal and conditional density function, respectively, and $\mathbf{J}_{g^\dagger}(\boldsymbol{z})$ is the Jacobian matrix of function $g^\dagger$ evaluated at $\boldsymbol{z}$. Based on Eq. (13), we have:

$$\mathbb{E}[Y|\boldsymbol{X}] = \int \boldsymbol{y} \cdot f_{Y|\boldsymbol{X}}(\boldsymbol{y}|\boldsymbol{x})dy = \int y \cdot f_{Y|\boldsymbol{Z}}(\boldsymbol{y}|\boldsymbol{z})dy = \mathbb{E}[Y|\boldsymbol{Z} = \boldsymbol{z}] = \mathbb{E}[Y|g(\boldsymbol{X}) = g(\boldsymbol{x})]. \quad (14)$$

$$\square$$

## A.2 PROOF OF COROLLARY 2.3.

*Proof.* For $\boldsymbol{X} = \boldsymbol{x}$, let $[\boldsymbol{c}||\boldsymbol{m}] \doteq [f_C^\dagger(\boldsymbol{x})||f_M^\dagger(\boldsymbol{x})] \doteq f^\dagger(\boldsymbol{x}) = \mathbf{A}^\dagger(\boldsymbol{x} - \boldsymbol{\alpha}_X)$, where $\mathbf{A}^\dagger$ is the left inverse of the full column-rank matrix $\mathbf{A}$ in Eq. (2), we have:

$$\begin{aligned}
CATE(\boldsymbol{x}) &= \mathbb{E}[Y|T = 1, \boldsymbol{C} = f_C^\dagger(\boldsymbol{x})] - \mathbb{E}[Y|T = 0, \boldsymbol{C} = f_C^\dagger(\boldsymbol{x})] \\
&= \mathbb{E}[Y|T = 1, \boldsymbol{C} = \boldsymbol{c}] - \mathbb{E}[Y|T = 0, \boldsymbol{C} = \boldsymbol{c}] \\
&= \mathbb{E}[\alpha_Y + \tau \cdot T + \sum \theta_j \cdot M_j + \sum \kappa_i \cdot C_i | T = 1, \boldsymbol{C} = \boldsymbol{c}] \\
&\quad - \mathbb{E}[\alpha_Y + \tau \cdot T + \sum \theta_j \cdot M_j + \sum \kappa_i \cdot C_i | T = 0, \boldsymbol{C} = \boldsymbol{c}] \\
&= \alpha_Y + \tau \cdot \mathbb{E}[T|T = 1, \boldsymbol{C} = \boldsymbol{c}] + \sum \theta_j \cdot \mathbb{E}[M_j|T = 1, \boldsymbol{C} = \boldsymbol{c}] + \sum \kappa_i \cdot \mathbb{E}[C_i|T = 1, \boldsymbol{C} = \boldsymbol{c}] \\
&\quad - \alpha_Y + \tau \cdot \mathbb{E}[T|T = 0, \boldsymbol{C} = \boldsymbol{c}] + \sum \theta_j \cdot \mathbb{E}[M_j|T = 0, \boldsymbol{C} = \boldsymbol{c}] + \sum \kappa_i \cdot \mathbb{E}[C_i|T = 0, \boldsymbol{C} = \boldsymbol{c}] \\
&= \tau \cdot (1 - 0) + \sum \theta_j \cdot (\gamma_j \cdot (1 - 0)) + \sum \kappa_i \cdot (c_i - c_i) \\
&= \tau + \sum \theta_j \cdot \gamma_j = \mathbb{E}[\tau + \sum \theta_j \cdot \gamma_j] = ATE,
\end{aligned}$$
$$(15)$$

where the first equality is due to the definition of CATE in Eq. (2). In addition, the causal estimand and bias of a proxy-of-confounder-based causal inference model that controls the latent variable $\boldsymbol{Z}$ inferred via $\hat{\boldsymbol{Z}} = \tilde{f}(\boldsymbol{X}) = \mathbf{B}^T \boldsymbol{X}$ (where $\mathbf{B}$ is also a full column-rank matrix) can be formulated as:

$$\begin{aligned}
DCEV(\mathbf{B}^T \boldsymbol{x}) &= \mathbb{E}[Y|T = 1, \hat{\boldsymbol{Z}} = \mathbf{B}^T \boldsymbol{x}] - \mathbb{E}[Y|T = 0, \hat{\boldsymbol{Z}} = \mathbf{B}^T \boldsymbol{x}] \\
&= \mathbb{E}[Y|T = 1, \hat{\boldsymbol{Z}} = \mathbf{B}^T \boldsymbol{\alpha}_X + \mathbf{B}^T \mathbf{A}[\boldsymbol{c}||\boldsymbol{m}]] - \mathbb{E}[Y|T = 0, \hat{\boldsymbol{Z}} = \mathbf{B}^T \boldsymbol{\alpha}_X + \mathbf{B}^T \mathbf{A}[\boldsymbol{c}||\boldsymbol{m}]] \\
&\overset{(a)}{=} \mathbb{E}[Y|T = 1, \boldsymbol{C} = \boldsymbol{c}, \boldsymbol{M} = \boldsymbol{m}] - \mathbb{E}[Y|T = 0, \boldsymbol{C} = \boldsymbol{c}, \boldsymbol{M} = \boldsymbol{m}] \\
&= \alpha_Y + \tau \cdot 1 + \sum \theta_j \cdot \mathbb{E}[M_j|T = 1, \boldsymbol{C} = \boldsymbol{c}, \boldsymbol{M} = \boldsymbol{m}] + \sum \kappa_i \cdot \mathbb{E}[C_i|T = 1, \boldsymbol{C} = \boldsymbol{c}, \boldsymbol{M} = \boldsymbol{m}] \\
&\quad - \alpha_Y + \tau \cdot 0 + \sum \theta_j \cdot \mathbb{E}[M_j|T = 0, \boldsymbol{C} = \boldsymbol{c}, \boldsymbol{M} = \boldsymbol{m}] + \sum \kappa_i \cdot \mathbb{E}[C_i|T = 0, \boldsymbol{C} = \boldsymbol{c}, \boldsymbol{M} = \boldsymbol{m}] \\
&= \tau \cdot (1 - 0) + \sum \theta_j \cdot (m_j - m_j) + \sum \kappa_i \cdot (c_i - c_i) \\
&= \tau = \mathbb{E}[\tau] = \mathbb{E}[DCEV(\mathbf{B}^T \boldsymbol{X})],
\end{aligned}$$
$$(16)$$

where step (a) is due to the fact that, since both $\mathbf{A}$ and $\mathbf{B}$ are full column-rank matrices, $\mathbf{B}^T \mathbf{A}$ is an invertible matrix, and the mapping $\bar{f} = \mathbf{B}^T \boldsymbol{\alpha}_X + \mathbf{B}^T \mathbf{A}$ is bijective. Therefore, we can invoke Lemma 2.1 and apply the left-inverse of $\bar{f}$, i.e., $\bar{f}^\dagger = (\mathbf{B}^T \mathbf{A})^{-1} - \mathbf{B}^T \boldsymbol{\alpha}_X$, to the condition of the expectation. The rest steps are based on the structural causal equations defined in Eq. (2). $\square$

### A.3 Another Case of Linear SCM with Latent Correlators

**Corollary A.1.** *(MixedCorrelator) For another Linear Structural Causal Model defined as follows:*

$$
\begin{aligned}
T &\leftarrow \mathbb{1}(\alpha_T + \sum \beta_i \cdot C_i > a) \\
M_j &\leftarrow \alpha_M + \gamma_j \cdot T + \phi_j \cdot U_j \\
\boldsymbol{X} &\leftarrow \boldsymbol{\alpha}_X + \mathbf{A}[\boldsymbol{M} \| \boldsymbol{C}] \\
Y &\leftarrow \alpha_Y + \tau \cdot T + \sum \theta_j \cdot U_j + \sum \kappa_i \cdot C_i,
\end{aligned}
\tag{17}
$$

*where the mixture function $f = \mathbf{A} \in \mathbb{R}^{K_X \times (K_C + K_M)}$ is a full column-rank matrix, the CATE, ATE, and the bias of proxy-of-confounder-based causal inference model that controls the latent variable $\hat{\boldsymbol{Z}}$ inferred via $\hat{\boldsymbol{Z}} = \tilde{f}(\boldsymbol{X}) = \mathbf{B}^T \boldsymbol{X}$ can be formulated as follows:*

$$
\begin{aligned}
ATE &= CATE = \tau \\
DEV(\hat{\boldsymbol{Z}}) &= \mathbb{E}[DCEV(\hat{\boldsymbol{Z}})] = DCEV(\hat{\boldsymbol{Z}}) = \tau - \sum \frac{\theta_j \cdot \gamma_j}{\phi_j} \\
Bias &= ATE - DEV(\mathbf{B}^T \boldsymbol{X}) = \sum \frac{\theta_j \cdot \gamma_j}{\phi_j},
\end{aligned}
\tag{18}
$$

*where $\mathbf{B} \in \mathbb{R}^{K_X \times (K_C + K_M)}$ is another full column-rank matrix. Since $\sum \frac{\theta_j \cdot \gamma_j}{\phi_j}$ is arbitrary, the estimator $DEV(\hat{\boldsymbol{Z}}) = \mathbb{E}[DCEV(\mathbf{B}^T \boldsymbol{X})]$ is arbitrarily biased for the estimation of ATE.*

*Proof.* The proof of the CATE and ATE is trivial. The causal estimand and the bias of a proxy-of-confounder-based causal inference model that controls the latent variables $\hat{\boldsymbol{Z}}$ inferred via $\hat{\boldsymbol{Z}} = \tilde{f}(\boldsymbol{X}) = \mathbf{B}^T \boldsymbol{X}$ (where $\mathbf{B}$ is also a full column-rank matrix) can be formulated as follows:

$$
\begin{aligned}
DCEV(\mathbf{B}^T \boldsymbol{x}) &= \mathbb{E}[Y|T = 1, \hat{\boldsymbol{Z}} = \mathbf{B}^T \boldsymbol{x}] - \mathbb{E}[Y|T = 0, \hat{\boldsymbol{Z}} = \mathbf{B}^T \boldsymbol{x}] \\
&= \mathbb{E}[Y|T = 1, \hat{\boldsymbol{Z}} = \boldsymbol{\alpha}_X + \mathbf{B}^T \mathbf{A}[\boldsymbol{c} \| \boldsymbol{m}]] - \mathbb{E}[Y|T = 0, \hat{\boldsymbol{Z}} = \boldsymbol{\alpha}_X + \mathbf{B}^T \mathbf{A}[\boldsymbol{c} \| \boldsymbol{m}]] \\
&\overset{(a)}{=} \mathbb{E}[Y|T = 1, \boldsymbol{C} = \boldsymbol{c}, \boldsymbol{M} = \boldsymbol{m}] - \mathbb{E}[Y|T = 0, \boldsymbol{C} = \boldsymbol{c}, \boldsymbol{M} = \boldsymbol{m}] \\
&= \alpha_Y + \tau \cdot 1 + \sum \theta_j \cdot \mathbb{E}[U_j|T = 1, \boldsymbol{C} = \boldsymbol{c}, \boldsymbol{M} = \boldsymbol{m}] + \sum \kappa_i \cdot \mathbb{E}[C_i|T = 1, \boldsymbol{C} = \boldsymbol{c}, \boldsymbol{M} = \boldsymbol{m}] \\
&\quad - \alpha_Y + \tau \cdot 0 + \sum \theta_j \cdot \mathbb{E}[U_j|T = 0, \boldsymbol{C} = \boldsymbol{c}, \boldsymbol{M} = \boldsymbol{m}] + \sum \kappa_i \cdot \mathbb{E}[C_i|T = 0, \boldsymbol{C} = \boldsymbol{c}, \boldsymbol{M} = \boldsymbol{m}] \\
&= \tau \cdot (1 - 0) + \sum \theta_j \cdot (\phi_j^{-1} \cdot (m_j - \alpha_M - \gamma_j) - \phi_j^{-1} \cdot (m_j - \alpha_M)) + \sum \kappa_i \cdot (c_i - c_i) \\
&= \tau - \sum \frac{\theta_j \cdot \gamma_j}{\phi_j} = \mathbb{E}\left[\tau - \sum \frac{\theta_j \cdot \gamma_j}{\phi_j}\right] = \mathbb{E}[DCEV(\mathbf{B}^T \boldsymbol{X})],
\end{aligned}
\tag{19}
$$
$\square$

where step (a) and the rest of the proof follow the same logic as the proof in Section 2.3.

### A.4 Proof of Theorem 3.1

The strict definitions of the exponential family, strong exponential (which is assumed for the factorized part of the conditional prior), and identifiability follow (Khemakhem et al., 2020; Lu et al., 2021), and can be referred to in Appendix E, F of (Lu et al., 2021), which we omit to avoid redundancy. The proof of Theorem 3.1 is largely based on the NF-iVAE paper (Lu et al., 2021), where most of the details can be found, with the new assumption introduced in CiVAE that each $\boldsymbol{S}_{f,i}$ has at least one invertible dimension incorporated to ensure that each dimension of the inferred latent variables is a bijective transformation of the corresponding true latent variable.

### A.4.1 PART I

**Step I**. In this step, we transform the equality of noisy conditional marginal distribution of $\boldsymbol{X}$ given $Y, T$ of two models with parameter $\boldsymbol{\theta}, \tilde{\boldsymbol{\theta}} \in \Theta$ into the equality of noise-free distributions.

$$
\begin{aligned}
& p_{\boldsymbol{\theta}}(\boldsymbol{X} \mid Y, T) = p_{\tilde{\boldsymbol{\theta}}}(\boldsymbol{X} \mid Y, T) \\
\Longrightarrow & \int_{\mathcal{Z}} p_f(\boldsymbol{X} \mid \boldsymbol{Z}) p_{\boldsymbol{S}, \boldsymbol{\lambda}}(\boldsymbol{Z} \mid Y, T) d\boldsymbol{Z} = \int_{\mathcal{Z}} p_{\tilde{f}}(\boldsymbol{X} \mid \boldsymbol{Z}) p_{\tilde{\boldsymbol{S}}, \tilde{\boldsymbol{\lambda}}}(\boldsymbol{Z} \mid Y, T) d\boldsymbol{Z} \\
\Longrightarrow & \int_{\mathcal{Z}} p_{\boldsymbol{\varepsilon}}(\boldsymbol{X} - f(\boldsymbol{Z})) p_{\boldsymbol{S}, \boldsymbol{\lambda}}(\boldsymbol{Z} \mid Y, T) d\boldsymbol{Z} = \int_{\mathcal{Z}} p_{\boldsymbol{\varepsilon}}(\boldsymbol{X} - \tilde{f}(\boldsymbol{Z})) p_{\tilde{\boldsymbol{S}}, \tilde{\boldsymbol{\lambda}}}(\boldsymbol{Z} \mid Y, T) d\boldsymbol{Z} \\
\stackrel{(a)}{\Longrightarrow} & \int_{\mathcal{X}} p_{\boldsymbol{\varepsilon}}(\boldsymbol{X} - \overline{\boldsymbol{X}}) p_{\boldsymbol{S}, \boldsymbol{\lambda}}\left(f^{\dagger}(\overline{\boldsymbol{X}}) \mid Y, T\right) \operatorname{vol}\left(\mathbf{J}_{f^{\dagger}}(\overline{\boldsymbol{X}})\right) d\overline{\boldsymbol{X}} = \\
& \int_{\mathcal{X}} p_{\boldsymbol{\varepsilon}}(\boldsymbol{X} - \overline{\boldsymbol{X}}) p_{\tilde{\boldsymbol{S}}, \tilde{\boldsymbol{\lambda}}}\left(\tilde{f}^{\dagger}(\overline{\boldsymbol{X}}) \mid Y, T\right) \operatorname{vol}\left(\mathbf{J}_{\tilde{f}^{\dagger}}(\overline{\boldsymbol{X}})\right) d\overline{\boldsymbol{X}} \\
\stackrel{(b)}{\Longrightarrow} & \int_{\mathbb{R}^d} p_{\boldsymbol{\varepsilon}}(\boldsymbol{X} - \overline{\boldsymbol{X}}) \tilde{p}_{f, \boldsymbol{S}, \boldsymbol{\lambda}, Y, T}(\overline{\boldsymbol{X}}) d\overline{\boldsymbol{X}} = \int_{\mathbb{R}^d} p_{\boldsymbol{\varepsilon}}(\boldsymbol{X} - \overline{\boldsymbol{X}}) \tilde{p}_{\tilde{f}, \tilde{\boldsymbol{S}}, \tilde{\boldsymbol{\lambda}}, \tilde{Y}, \tilde{T}}(\overline{\boldsymbol{X}}) d\overline{\boldsymbol{X}} \\
\Longrightarrow & \left(\tilde{p}_{f, \boldsymbol{S}, \boldsymbol{\lambda}, Y, T} * p_{\boldsymbol{\varepsilon}}\right)(\boldsymbol{X}) = \left(\tilde{p}_{\tilde{f}, \tilde{\boldsymbol{S}}, \tilde{\boldsymbol{\lambda}}, \tilde{Y}, \tilde{T}} * p_{\boldsymbol{\varepsilon}}\right)(\boldsymbol{X}) \\
\stackrel{(c)}{\Longrightarrow} & F\left[\tilde{p}_{f, \boldsymbol{S}, \boldsymbol{\lambda}, Y, T}\right](\boldsymbol{\omega}) \varphi_{\boldsymbol{\varepsilon}}(\boldsymbol{\omega}) = F\left[\tilde{p}_{\tilde{f}, \tilde{\boldsymbol{S}}, \tilde{\boldsymbol{\lambda}}, \tilde{Y}, \tilde{T}}\right](\boldsymbol{\omega}) \varphi_{\boldsymbol{\varepsilon}}(\boldsymbol{\omega}) \\
\stackrel{(d)}{\Longrightarrow} & F\left[\tilde{p}_{f, \boldsymbol{S}, \boldsymbol{\lambda}, Y, T}\right](\boldsymbol{\omega}) = F\left[\tilde{p}_{\tilde{f}, \tilde{\boldsymbol{S}}, \tilde{\boldsymbol{\lambda}}, \tilde{Y}, \tilde{T}}\right](\boldsymbol{\omega}) \\
\Longrightarrow & \tilde{p}_{f, \boldsymbol{S}, \boldsymbol{\lambda}, Y, T}(\boldsymbol{X}) = \tilde{p}_{\tilde{f}, \tilde{\boldsymbol{S}}, \tilde{\boldsymbol{\lambda}}, \tilde{Y}, \tilde{T}}(\boldsymbol{X}).
\end{aligned}
\tag{20}
$$

Step (a) is based on the rule of change-of-variable, where $\operatorname{vol}(\mathbf{A}) = \sqrt{\det\left(\mathbf{A}^T \mathbf{A}\right)}$. In step (b), we define $\tilde{p}_{f, \boldsymbol{S}, \boldsymbol{\lambda}, Y, T}(\boldsymbol{X}) \triangleq p_{\boldsymbol{S}, \boldsymbol{\lambda}}\left(f^{\dagger}(\boldsymbol{X}) \mid Y, T\right) \operatorname{vol}\left(\mathbf{J}_{f^{\dagger}}(\boldsymbol{X})\right) \mathbb{I}_{\mathcal{X}}(\boldsymbol{X})$. In step (c), we use $F[\cdot]$ to denote the Fourier transform. In step (d), we drop $\varphi_{\boldsymbol{\varepsilon}}(\boldsymbol{\omega})$ as it is non-zero *a.e.* (see Assumption 3).

**Step II**. In this step, we transform the equality of the noise-free distributions into the relationship of the sufficient statistics $\boldsymbol{S}$ and $\tilde{\boldsymbol{S}}$. By taking logarithm of both sides of Eq. (20), we have:

$$
\begin{aligned}
& \log \operatorname{vol}\left(J_{f^{\dagger}}(\boldsymbol{X})\right) + \log \mathcal{Q}\left(f^{\dagger}(\boldsymbol{X})\right) - \log \mathcal{C}(Y, T) + \left\langle \boldsymbol{S}\left(f^{\dagger}(\boldsymbol{X})\right), \boldsymbol{\lambda}(Y, T)\right\rangle \\
& = \log \operatorname{vol}\left(J_{\tilde{f}^{\dagger}}(\boldsymbol{X})\right) + \log \tilde{\mathcal{Q}}\left(\tilde{f}^{\dagger}(\boldsymbol{X})\right) - \log \tilde{\mathcal{C}}(Y, T) + \left\langle \tilde{\boldsymbol{S}}\left(\tilde{f}^{\dagger}(\boldsymbol{X})\right), \tilde{\boldsymbol{\lambda}}(Y, T)\right\rangle.
\end{aligned}
\tag{21}
$$

Let $(Y, T)_0, \cdots, (Y, T)_k$ be the $k + 1$ distinct points defined in Assumption 3 - (iv). We obtain $k + 1$ equations by evaluating the Eq. (21) at these points, where the first equation is subtracted from the remaining ones, which leads to the following equation system:

$$
\begin{aligned}
& \left\langle \boldsymbol{S}\left(f^{\dagger}(\boldsymbol{X})\right), \boldsymbol{\lambda}((Y, T)_l) - \boldsymbol{\lambda}((Y, T)_0)\right\rangle + \log \frac{\mathcal{C}((Y, T)_0)}{\mathcal{C}((Y, T)_l)} \\
& = \left\langle \tilde{\boldsymbol{S}}\left(\tilde{f}^{\dagger}(\boldsymbol{X})\right), \tilde{\boldsymbol{\lambda}}((Y, T)_l) - \tilde{\boldsymbol{\lambda}}((Y, T)_0)\right\rangle + \log \frac{\tilde{\mathcal{C}}((Y, T)_0)}{\tilde{\mathcal{C}}((Y, T)_l)}, \quad l = 1, \cdots, k.
\end{aligned}
\tag{22}
$$

Let $\mathbf{L}$ be the invertible matrix defined in Assumption 3 - (iv) and $\tilde{\mathbf{L}}$ be the counterpart for $\tilde{\boldsymbol{\lambda}}$, if we summarize all terms irrelevant to $\boldsymbol{X}$ into a constant $\boldsymbol{b}$, we have:

$$
\begin{aligned}
& \mathbf{L}^T \boldsymbol{S}\left(f^{\dagger}(\boldsymbol{X})\right) = \tilde{\mathbf{L}}^T \tilde{\boldsymbol{S}}\left(\tilde{f}^{\dagger}(\boldsymbol{X})\right) + \boldsymbol{b} \\
\Longrightarrow & \boldsymbol{S}\left(f^{\dagger}(\boldsymbol{X})\right) = \mathbf{A} \tilde{\boldsymbol{S}}\left(\tilde{f}^{\dagger}(\boldsymbol{X})\right) + \boldsymbol{c},
\end{aligned}
\tag{23}
$$

where $\mathbf{A} = \mathbf{L}^{-T} \tilde{\mathbf{L}} \in \mathbb{R}^{k \times k}$, and $\boldsymbol{c} = \mathbf{L}^{-T} \boldsymbol{b} \in \mathbb{R}^k$.

**Step III**. Ideally, to prove the element-wise bijective identifiability of the latent variables $\boldsymbol{Z}$, the transformation of the sufficient statistics $\boldsymbol{S}$ derived in Eq. (23) should be bijective. We claim that if the conditional prior $p_{\boldsymbol{S}, \boldsymbol{\lambda}}(\boldsymbol{Z} \mid Y, T)$ is strongly exponential and $\mathbf{L}$ is invertible, $\tilde{\mathbf{L}}$ and $\mathbf{A}$ must also be invertible. The proof is omitted, and can be referred to in Appendix H.1.1 of (Lu et al., 2021).

### A.4.2 PART II

In this part, we prove that, if Assumptions 1, 2 and 3 hold, we can identify the factorized part of the sufficient statistics $\boldsymbol{S}(\boldsymbol{Z})$, i.e., $\boldsymbol{S}_f(\boldsymbol{Z})$, up to permutation and element-wise transformation. Specifically, if we use $\boldsymbol{v}$ to denote the composite map $\tilde{f}^\dagger \circ f : \mathcal{Z} \to \mathcal{Z}$, Eq. (23) can be rewritten into:

$$\boldsymbol{S}(\boldsymbol{Z}) = \mathbf{A}\tilde{\boldsymbol{S}}(\boldsymbol{v}(\boldsymbol{Z})) + \boldsymbol{c}. \tag{24}$$

We aim to prove that $\mathbf{A}$ in Eq. (24) is a block permutation matrix.

**Step I**. We start by showing that $\boldsymbol{v}$ is a component-wise function. If we differentiate both sides of Eq. (24) with respect to $Z_s$ and $Z_t$, where $s \neq t$, we have:

$$\frac{\partial \boldsymbol{S}(\boldsymbol{Z})}{\partial Z_s} = \mathbf{A}\sum_{i=1}^{K_Z} \frac{\partial \tilde{\boldsymbol{S}}(\boldsymbol{v}(\boldsymbol{Z}))}{\partial v_i(\boldsymbol{Z})} \cdot \frac{\partial v_i(\boldsymbol{Z})}{\partial Z_s}$$

$$\frac{\partial^2 \boldsymbol{S}(\boldsymbol{Z})}{\partial Z_s \partial Z_t} = \mathbf{A}\sum_{i=1}^{K_Z} \sum_{i=1}^{K_Z} \frac{\partial^2 \tilde{\boldsymbol{S}}(\boldsymbol{v}(\boldsymbol{Z}))}{\partial v_i(\boldsymbol{Z}) \partial v_j(\boldsymbol{Z})} \cdot \frac{\partial v_j(\boldsymbol{Z})}{\partial Z_t} \cdot \frac{\partial v_i(\boldsymbol{Z})}{\partial Z_s} + \mathbf{A}\sum_{i=1}^{K_Z} \frac{\partial \tilde{\boldsymbol{S}}(\boldsymbol{v}(\boldsymbol{Z}))}{\partial v_i(\boldsymbol{Z})} \cdot \frac{\partial^2 v_i(\boldsymbol{Z})}{\partial Z_s \partial Z_t}. \tag{25}$$

Note that for the factorized part of the sufficient statistics $\boldsymbol{S}$, i.e., $\boldsymbol{S}_f$, all *cross-derivatives* are zero, and for the non-factorized part of $\boldsymbol{S}$, i.e., $\boldsymbol{S}_{nf}$, which is a neural network with ReLU activation (i.e., linear *a.e.*), all *second-order derivatives* are zero. Therefore, the *second order cross-derivatives* on the LHS. of Eq. (25) are zero, which leads to the following equality:

$$\mathbf{0} = \mathbf{A}\sum_{i=1}^{K_Z} \frac{\partial^2 \tilde{\boldsymbol{S}}(\boldsymbol{v}(\boldsymbol{Z}))}{\partial v_i(\boldsymbol{Z})^2} \cdot \frac{\partial v_i(\boldsymbol{Z})}{\partial Z_t} \cdot \frac{\partial v_i(\boldsymbol{Z})}{\partial Z_s} + \mathbf{A}\sum_{i=1}^{K_Z} \frac{\partial \tilde{\boldsymbol{S}}(\boldsymbol{v}(\boldsymbol{Z}))}{\partial v_i(\boldsymbol{Z})} \cdot \frac{\partial^2 v_i(\boldsymbol{Z})}{\partial Z_s \partial Z_t}. \tag{26}$$

Eq. (26) can be written into the matrix-vector product form as follows:

$$\mathbf{0} = \mathbf{A}\tilde{\boldsymbol{S}}''(\boldsymbol{Z})\boldsymbol{v}'_{s,t}(\boldsymbol{Z}) + \mathbf{A}\tilde{\boldsymbol{S}}'(\boldsymbol{Z})\boldsymbol{v}''_{s,t}(\boldsymbol{Z}), \tag{27}$$

where

$$\tilde{\boldsymbol{S}}''(\boldsymbol{Z}) = \left[ \frac{\partial^2 \tilde{\boldsymbol{S}}(\boldsymbol{v}(\boldsymbol{Z}))}{\partial v_1(\boldsymbol{Z})^2}, \cdots, \frac{\partial^2 \tilde{\boldsymbol{S}}(\boldsymbol{v}(\boldsymbol{Z}))}{\partial v_{K_Z}(\boldsymbol{Z})^2} \right] \in \mathbb{R}^{k \times K_Z},$$

$$\boldsymbol{v}'_{s,t}(\boldsymbol{Z}) = \left[ \frac{\partial v_1(\boldsymbol{Z})}{\partial Z_t} \cdot \frac{\partial v_1(\boldsymbol{Z})}{\partial Z_s}, \cdots, \frac{\partial v_{K_Z}(\boldsymbol{Z})}{\partial Z_t} \cdot \frac{\partial v_{K_Z}(\boldsymbol{Z})}{\partial Z_s} \right]^T \in \mathbb{R}^{K_Z},$$

and

$$\tilde{\boldsymbol{S}}'(\boldsymbol{Z}) = \left[ \frac{\partial \tilde{\boldsymbol{S}}(\boldsymbol{v}(\boldsymbol{Z}))}{\partial v_1(\boldsymbol{Z})}, \cdots, \frac{\partial \tilde{\boldsymbol{S}}(\boldsymbol{v}(\boldsymbol{Z}))}{\partial v_{K_Z}(\boldsymbol{Z})} \right] \in \mathbb{R}^{k \times K_Z},$$

$$\boldsymbol{v}''_{s,t}(\boldsymbol{Z}) = \left[ \frac{\partial^2 v_1(\boldsymbol{Z})}{\partial Z_s \partial Z_t}, \cdots, \frac{\partial^2 v_{K_Z}(\boldsymbol{Z})}{\partial Z_s \partial Z_t} \right]^T \in \mathbb{R}^{K_Z}.$$

If we denote the concatenation as $\tilde{\boldsymbol{S}}'''(\boldsymbol{Z}) = \left[ \tilde{\boldsymbol{S}}''(\boldsymbol{Z}), \tilde{\boldsymbol{S}}'(\boldsymbol{Z}) \right] \in \mathbb{R}^{k \times 2K_Z}$ and $\boldsymbol{v}'''_{s,t}(\boldsymbol{Z}) = \left[ \boldsymbol{v}'_{s,t}(\boldsymbol{Z})^T, \boldsymbol{v}''_{s,t}(\boldsymbol{Z})^T \right]^T \in \mathbb{R}^{2K_z}$, we have:

$$\mathbf{0} = \mathbf{A}\tilde{\boldsymbol{S}}'''(\boldsymbol{Z})\boldsymbol{v}'''_{s,t}(\boldsymbol{Z}). \tag{28}$$

Finally, if we denote the rows of $\tilde{\boldsymbol{S}}'''(\boldsymbol{Z})$ that correspond to the factorized part of $\boldsymbol{S}$ by $\tilde{\boldsymbol{S}}'''_f(\boldsymbol{Z})$, according to Lemma 5 of the iVAE paper (Khemakhem et al., 2020) and the assumption that $k \geq 2K_Z$, we have that the rank of $\tilde{\boldsymbol{S}}'''_f(\boldsymbol{Z})$ is $2K_Z$. Since $k \geq 2K_Z$, the rank of $\tilde{\boldsymbol{S}}'''(\boldsymbol{Z})$ is also $2K_Z$. Since the rank of $\mathbf{A}$ is $k$, the rank of $\mathbf{A}\tilde{\boldsymbol{S}}'''(\boldsymbol{Z})$ is $2K_Z$, which implies that $\boldsymbol{v}'''_{s,t}(\boldsymbol{Z}) \in \mathbb{R}^{2K_Z}$ is a zero vector. Therefore, we have $\boldsymbol{v}'_{s,t}(\boldsymbol{Z}) = \mathbf{0}, \forall s \neq t$, and we have demonstrated that $\boldsymbol{v}$ is a component-wise function.

**Step II**. Based on **Step I**, we demonstrate that $\mathbf{A}$ is a block permutation matrix. Without loss of generality, we assume that the permutation in $\boldsymbol{v}$ is Identity, where $\boldsymbol{v}(\boldsymbol{Z}) = [v_1(Z_1), \cdots, v_{K_Z}(Z_{K_Z})]^T$ and each $v_i$ is a nonlinear univariate scalar function. Since $f$ and $\tilde{f}$ are injective, $\boldsymbol{v}$ is bijective and

$\boldsymbol{v}^{-1}(\boldsymbol{Z}) = \left[ v_1^{-1}(Z_1), \cdots, v_{K_Z}^{-1}(Z_{K_Z}) \right]^T$. If we denote $\overline{\boldsymbol{S}}(\boldsymbol{v}(\boldsymbol{Z})) = \tilde{\boldsymbol{S}}(\boldsymbol{v}(\boldsymbol{Z})) + \mathbf{A}^{-1}\boldsymbol{c}$, Eq. (24) can be reformulated as $\boldsymbol{S}(\boldsymbol{Z}) = \mathbf{A}\overline{\boldsymbol{S}}(\boldsymbol{v}(\boldsymbol{Z}))$. We then apply $\boldsymbol{v}^{-1}$ to $\boldsymbol{Z}$ on both sides, which gives

$$S\left(\boldsymbol{v}^{-1}(\boldsymbol{Z})\right) = \mathbf{A}\overline{\boldsymbol{S}}(\boldsymbol{Z}). \tag{29}$$

Let $t$ be the index of an entry in $\boldsymbol{S}$ that corresponds to the factorized part $\boldsymbol{S}_f$. For all $s \neq t$, we have:

$$0 = \frac{\partial \boldsymbol{S}\left(\boldsymbol{v}^{-1}(\boldsymbol{Z})\right)_t}{\partial Z_s} = \sum_{j=1}^{k} a_{tj} \frac{\partial \overline{\boldsymbol{S}}(\boldsymbol{Z})_j}{\partial Z_s}. \tag{30}$$

Since the entries of $\tilde{\boldsymbol{S}}$ are linearly independent, $a_{tj}$ is zero for any $j$ such that $\frac{\partial \overline{\boldsymbol{S}}(\boldsymbol{Z})_j}{\partial Z_s} \neq 0$. This includes the entries $S_j$ that correspond to **1)** the factorized part that does not depend on $Z_t$; and **2)** the non-factorized part $\boldsymbol{S}_{nf}$. Therefore, when $t$ is the index of an entry in the sufficient statistics $\boldsymbol{S}$ that corresponds to factor $i$ in the factorized part $\boldsymbol{S}_f$, i.e., $\boldsymbol{S}_{f,i}$, the only non-zero $a_{tj}$ are the ones that map between $\boldsymbol{S}_{f,i}(Z_i)$ and $\overline{\boldsymbol{S}}_{f,i}(v_i(Z_i))$. Therefore, we can construct an invertible submatrix $\mathbf{A}'_i$ with all non-zero elements $a_{tj}$ for all $t$ that corresponds to factor $i$, such that

$$\boldsymbol{S}_{f,i}(Z_i) = \mathbf{A}'_i \overline{\boldsymbol{S}}_{f,i}(v_i(Z_i)) = \mathbf{A}'_i \tilde{\boldsymbol{S}}_{f,i}(v_i(Z_i)) + \boldsymbol{c}_i, \quad i = 1, \cdots, K_Z, \tag{31}$$

where $\boldsymbol{c}_i$ denotes the corresponding elements of $\boldsymbol{c}$. Eq. (31) means that for each $i = 1, \cdots, K_Z$, the matrix block $\mathbf{A}'_i$ of $\mathbf{A}$ affinely transforms the $i$-specific sufficient statistics vector $\boldsymbol{S}_{f,i}(Z_i)$ into $\tilde{\boldsymbol{S}}_{f,i}(v_i(Z_i))$. In addition, there is also an additional block $\mathbf{A}'$ that affinely transforms $\boldsymbol{S}_{nf}(\boldsymbol{Z})$ in into $\boldsymbol{S}_{nf}(v(\boldsymbol{Z}))$. This completes the proof that $\mathbf{A}$ is a block permutation matrix.

### A.4.3 PART III

Let $\tilde{Z}_i = v_i(Z_i) = \tilde{f}^\dagger(\boldsymbol{X})_i$ be the $i$th inferred latent variable. Assume again that the permutation in $\boldsymbol{v}$ is Identity. In this part, we prove that if Assumption 2 holds, each inferred latent variable $\tilde{Z}_i$ is the bijective transformation of the true latent variable. The proof is as follows.

*Proof.* Plugging $\tilde{Z}_i$ into Eq. (31), we have:

$$\boldsymbol{S}_{f,i}(Z_i) = \mathbf{A}'_i \bar{\boldsymbol{S}}_{f,i}(\tilde{Z}_i). \tag{32}$$

According to Assumption 2, there exists one dimension of $\boldsymbol{S}_{f,i}$, i.e., $j$, such that $S_{f,ij}$ is bijective. This implies that $\boldsymbol{S}_{f,i}$ is injective, and therefore it has a left-inverse $\boldsymbol{S}^\dagger_{f,i}$. we apply $\boldsymbol{S}^\dagger_{f,i}$ to both sides of Eq. (32), which gives:

$$Z_i = \boldsymbol{S}^\dagger_{f,i} \mathbf{A}'_i \bar{\boldsymbol{S}}_{f,i}(\tilde{Z}_i). \tag{33}$$

Since $\mathbf{A}'_i$ is a block of an invertible block permutation matrix, $\mathbf{A}_i$ is also an invertible matrix, and therefore $\mathbf{A}'_i$ is a bijective mapping. In addition, since $\tilde{\boldsymbol{S}}_{f,i}$ is injective, $\bar{\boldsymbol{S}}_{f,i}$ is also injective, and therefore the composite map $\boldsymbol{S}^\dagger_{f,i} \mathbf{A}'_i \bar{\boldsymbol{S}}_{f,i} : \mathbb{R} \to \mathbb{R}$ that applies on $\tilde{Z}_i$ is a bijective. This completes the proof that each inferred latent variable $\tilde{Z}_i$ is the bijective transformation of the true latent variable in the case of no noise, where $\boldsymbol{Z} = f^\dagger(\boldsymbol{X})$ are the true latent variables. If noise $\varepsilon$ exists, the posterior distribution of the latent variables can be identified up to an analogous bijective indeterminacy. $\square$

### A.4.4 CONSISTENCY

*Proof.* If the family of the variational posterior $q_\phi(\boldsymbol{Z}|\boldsymbol{X}, Y, T)$ contains the true posterior $p_\theta(\boldsymbol{Z}|\boldsymbol{X}, Y, T)$, then by optimizing the loss of Eq. (10) (with the KL term replaced by the score matching loss defined in Eq. (11)) over its parameter $\phi$, the score matching term will eventually vanish. Therefore, the ELBO term in Eq. (10) will be equal to the log-likelihood. Under this circumstance, CiVAE inherits all the properties of maximum likelihood estimation (MLE). Since the identifiability of CiVAE is guaranteed up to permutation and component-wise bijective transformation of the latent variables, the consistency property of MLE means that the model will converge to the true parameter $\boldsymbol{\theta}^*$ up to such mild indeterminacy of the latent variables in the limit of infinite data. $\square$

A.5 PROOF OF THEOREM 3.2

*Proof.* Let $\boldsymbol{C}$ be the true latent confounders and $\hat{\boldsymbol{C}}$ be the transformed confounders, where the transformation function $g$ is bijective and differentiable *a.e.* Let $g^{-1}$ denote its inverse. The ATE estimator that controls transformed confounders $\hat{\boldsymbol{C}}$ can be formulated as:

$$DEV(\hat{\boldsymbol{C}}) = \mathbb{E}_{p(\hat{\boldsymbol{C}})}[\mathbb{E}[Y|T = 1, \hat{\boldsymbol{C}} = \hat{\boldsymbol{c}}] - \mathbb{E}[Y|T = 0, \hat{\boldsymbol{C}} = \hat{\boldsymbol{c}}]]. \tag{34}$$

Specifically, for the continuous case where density functions exist, for each term, we have:

$$\mathbb{E}_{p(\hat{\boldsymbol{C}})}[\mathbb{E}[Y|T = t, \hat{\boldsymbol{C}} = \hat{\boldsymbol{c}}]] = \int f_{\hat{\boldsymbol{C}}}(\hat{\boldsymbol{c}}) \int y \cdot f_{Y|T,\hat{\boldsymbol{C}}}(y|t, \hat{\boldsymbol{c}}) dy d\hat{\boldsymbol{c}}. \tag{35}$$

For the marginal density $f_{\hat{\boldsymbol{C}}}(\hat{\boldsymbol{c}})$, the following equality holds:

$$f_{\hat{\boldsymbol{C}}}(\hat{\boldsymbol{c}}) = f_{\boldsymbol{C}}(g^{-1}(\hat{\boldsymbol{c}}))|J_{g^{-1}}(\hat{\boldsymbol{c}})| = f_{\boldsymbol{C}}(\boldsymbol{c})|J_{g^{-1}}(\hat{\boldsymbol{c}})|. \tag{36}$$

As for the conditional density $f_{Y|T,\hat{\boldsymbol{C}}}(y|t, \hat{\boldsymbol{c}})$, since $g$ is bijective, according to Eq. (13), we have:

$$f_{Y|T,\hat{\boldsymbol{C}}}(y|t, \hat{\boldsymbol{c}}) = f_{Y|T,\boldsymbol{C}}(y|t, \boldsymbol{c}). \tag{37}$$

Combining Eqs. (36) and (37), and given that $d\hat{\boldsymbol{c}} = |J_g(\boldsymbol{c})|d\boldsymbol{c}$, we have:

$$\begin{aligned}
(35) &= \int f_{\boldsymbol{C}}(\boldsymbol{c})|\mathbf{J}_{g^{-1}}(\hat{\boldsymbol{c}})| \int y \cdot f_{Y|T,\boldsymbol{C}}(y|t, \boldsymbol{c}) dy |\mathbf{J}_g(\boldsymbol{c})| d\boldsymbol{c} \\
&= |\mathbf{J}_{g^{-1}}(\hat{\boldsymbol{c}})| \cdot |\mathbf{J}_g(\boldsymbol{c})| \int f_{\boldsymbol{C}}(\boldsymbol{c}) \int y \cdot f_{Y|T,\boldsymbol{C}}(y|t, \boldsymbol{c}) dy d\boldsymbol{c} \\
&\overset{(a)}{=} \int f_{\boldsymbol{C}}(\boldsymbol{c}) \int y \cdot f_{Y|T,\boldsymbol{C}}(y|t, \boldsymbol{c}) dy d\boldsymbol{c} \\
&= \mathbb{E}_{p(\boldsymbol{C})}[\mathbb{E}[Y|T = t, \boldsymbol{C} = \boldsymbol{c}]],
\end{aligned} \tag{38}$$

where the term $|J_{g^{-1}}(\hat{\boldsymbol{c}})| \cdot |J_g(\boldsymbol{c})|$ vanishes in step (a) as the two factors have the product of one. Therefore, if we plug Eq. (38) into Eq. (34), it leads to the following equality:

$$\begin{aligned}
DEV(\hat{\boldsymbol{C}}) &= \mathbb{E}_{p(\hat{\boldsymbol{C}})}[\mathbb{E}[Y|T = 1, \hat{\boldsymbol{C}} = \hat{\boldsymbol{c}}] - \mathbb{E}[Y|T = 0, \hat{\boldsymbol{C}} = \hat{\boldsymbol{c}}]] \\
&= \mathbb{E}_{p(\boldsymbol{C})}[\mathbb{E}[Y|T = 1, \boldsymbol{C} = \boldsymbol{c}] - \mathbb{E}[Y|T = 0, \boldsymbol{C} = \boldsymbol{c}]] \\
&= DEV(\boldsymbol{C}) = ATE,
\end{aligned} \tag{39}$$

where the last step is due to Eq. (2) in Definition 2, which completes our proof that controlling bijectively transformed confounders provides an unbiased estimation of ATE. □

