# OpenReview forum: "Causal Effect Estimation with Mixed Latent Confounders and Post-treatment Variables"
_ICLR.cc/2024/Conference — ICLR 2024 Conference Withdrawn Submission_

### Official Review · Reviewer_cihL · 2023-10-27

**Soundness:** 2 fair
**Presentation:** 2 fair
**Contribution:** 2 fair
**Rating:** 3
**Confidence:** 3

**Summary:**

In the cases of only post-treatment proxies available, this paper proposes a latent post-treatment variable causal graph. Based on the causal graph, the authors use deep variational inference to learn latent representation and then use conditional dependence properties to distinguish latent confounders and latent post-treatment variables.

**Strengths:**

- The authors study the issue of latent post-treatment variables, which might be vital in some specific scenarios.
- The core idea is clearly explained and easy to follow.

**Weaknesses:**

**[Algorithm]** Since the authors have modeled the potential causal graph (Figure 1(c)), why not directly use the causal relationships on the graph to model variables C and M using VAE, instead of modeling variable Z first and then using causal discovery algorithms to distinguish them?

**[Major Mistakes in Algorithm]** In Section 3.4, this paper ignores the direct causal relationship between $C_i$ and $C_j$, i.e., $C_i  \rightarrow C_j$ or $C_i  \leftarrow C_j$, and also overlooks the direct causal relationship between $M_i$ and $M_j$, i.e., $M_i  \rightarrow M_j$ or $M_i  \leftarrow M_j$. For instance, in the case of $\\{M_i \rightarrow T \leftarrow M_j, M_i  \rightarrow M_j\\}$, the direct causal effect of $M_i  \rightarrow M_j$ may counteract the direct causal effect of $M_i \rightarrow T \leftarrow M_j$, resulting in $M_i$ and $M_j$ being marginally independent, but they become dependent after conditioning on the assigned treatment T. The proposed algorithm exhibits theoretical flaws, thereby casting doubt on the credibility of this paper. Why don't the authors opt for existing causal discovery algorithms to identify causal relationships? Furthermore, this paper neglects the essential principles of faithfulness and the causal Markov assumption required for causal discovery.

**[Strong Assumption]** In assumption 1, the assumption that $f(C,M)$ is an injective function is quite strong. Even linear models like $X=C+M$ do not satisfy this assumption. For nonlinear functions, it is more difficult to meet this assumption in practical scenarios. Pearl, (2012), Rothman et al., (2008), and Louizos et al., (2017) do not implicitly make the **Noisy-Injectivity** assumption, but rather adopt some weaker assumptions, such as **X** having at least the same number of categories as **C**. The **Noisy-Injectivity** assumption used in this paper is a stronger and more difficult assumption to satisfy. Besides, does this paper require the dimension of post-treatment proxies X to be larger than the sum of the dimensions of C and M, specifically in linear models?

**[Post-treatment proxies]** Why can't we directly collect pre-treatment proxies, instead of choosing post-treatment proxies to identify causal effects? This is different from the traditional observational settings in causal literature, which makes me doubt the rationale of causal graphs. Can the author provide causal literature that supports the use of post-treatment proxies? Because the examples provided by the author are really difficult to make sense of.

**[Setting]** Although the problem discussed in this paper is interesting, it is highly questionable whether such scenarios exist in the real world. Because people typically record information based on the chronological order of events, and practitioners would choose treatments based on these pre-treatment variables. In which scenarios are pre-treatment proxies and post-treatment proxies indistinguishable? Additionally, why are there no edges from pre-treatment variables to post-treatment variables in the Causal Graph in Figure 1(c)? This assumption is quite strong. Does such a causal relationship truly exist in actual scenarios?

**[Confusing Example]** The real-world example provided in this paper is quite confusing. It is apparent that variables like age, gender, and geographical location would direct affect the decision to transition from onsite to online work. The studied effect in this example seems an anti-causal problem. Additionally, the qualifications required for an online job are application requirements, rather than applicants' covariates. Furthermore, I am curious about when the applicants' covariates were collected. Shouldn't the company make their decision to switch to online work based on these covariates? In that case, wouldn't the pre-treatment already be distinguishable? The author should offer a more concrete example.  This leaves me very confused about whether the setting of this article is valid, because as far as I know, pre-treatment variables and post-treatment variables can be easily distinguished based on the time of data collection, and the sequence of events is also readily available in most cases.

**[Experiments]** [A Critical Look at the Consistency of Causal Estimation with Deep Latent Variable Models, NeuIPS 2021] demonstrated that CEVAE could only solve simple case problems. If the proxy variables only contain pre-treatment variable information or only post-treatment variable information, i.e., the true causal relationships violated the assumed causal graph in Figure 1(c), will the proposed method still perform well? The authors should provide more experiments to demonstrate this.  Additionally, can the proposed method handle high-dimensional proxy scenarios?

**Typos**: *DCVE*(**X**) → *DCEV*(**X**)

**Questions:**

See Above.

---

### Official Review · Reviewer_gqDy · 2023-10-28

**Soundness:** 3 good
**Presentation:** 3 good
**Contribution:** 2 fair
**Rating:** 5
**Confidence:** 3

**Summary:**

The paper considers a causal inference setting where the observed covariates are proxies of unobserved confounders and post-treatment variables. First, a bias-formula is derived which shows that standard latent variable models that control for the observed covariates are biased due to conditioning on post-treatment variables. Then, the authors propose a new variational autoencoder-type latent variable model that disentangles the latent unobserved confounders from the latent post-treatment variables and allows for unbiased estimation. Identifiability guarantees are derived under assumptions on the data-generating process and the latent variable distributions. Finally, the authors show empirically that their method outperforms latent-variable models from literature.

**Strengths:**

- The paper is well written and theoretically sounds, proofs for all statements are provided in the appendix
- The proposed method performs well empirically

**Weaknesses:**

- I am not fully convinced of the relevance of the problem setting. The paper assumes that the observed covariates $X$ are (at least part) post-treatment variables. I am not aware of any practical examples where post-treatment variables are used as control variables for causal inference (for the exact reason that bias by conditioning on post-treatment variables is a well-known problem). Practicioners only control for pre-treatment covariates in order to exclude this type of bias. I like the introductory example as a motivation (on-side/remote jobs), but I think it would be helpful to provide some examples/references where such a type of analysis is actually applied.
- In order to provide identifiability guarantees, the paper relies on strong and untestable assumptions (e.g., injective mapping between latent factors and proxies), which may be hard to verify even with domain knowledge
- No details regarding implementation, hyperparameter tuning, the real-world dataset and baselines are provided. Furthermore, the Code has not been made available.
- I think the paper would benefitt from a case-study on real-world data. If I understand correctly, ony synthetic and semi-synthetic data have been used in the experiments

**Questions:**

- In the bias formula of Theorem 2.2., should the last minus be a plus? Since you are subtracting a difference ($DEV$).

---

### Official Review · Reviewer_Eaxa · 2023-10-31

**Soundness:** 3 good
**Presentation:** 2 fair
**Contribution:** 2 fair
**Rating:** 3
**Confidence:** 4

**Summary:**

This paper studied a critical problem by considering the covariates with both confounders and post-treatment variables in causal inference. The proposed method has a well-established theoretical analysis.

**Strengths:**

1. The studied problem, i.e., covariates with both confounders and post-treatment variables, is important and practical.
2. The proposed method has a well-established theoretical analysis

**Weaknesses:**

1.	Recent advances have analyzed more general cases from theoretical and algorithmic perspectives [1,2]. To be specific, such problems consider the existence of both treatment-correlated (containing post-treatment variables) and outcome-correlated variables. Hence, I think the contribution of the paper is limited compared to previous work in this community.
2.	Meanwhile, I have to point out that this paper has omitted a wide of related work in the covariate disentanglement area. Either in the related work section or in the experimental part.
3.	Besides, for variable decomposition, semi-parametric theory can already offer solid solutions [1,2].
4.	I think that either analyzing the effect of the existence of post-treatment variables or using the iVAE framework are not novel contributions. The former is very common for statistical analysis in linear models (bias caused by conditioning on mediators), while the latter (identification for conditional exponential families up to linear, point-wise statistical transformations) is not new to me.

[1] Andrea Rotnitzky and Ezequiel Smucler. 2020. Efficient Adjustment Sets for Population Average Causal Treatment Effect Estimation in Graphical Models. J. Mach. Learn. Res. 21, 188 (2020), 1–86
[2] Wang H, Kuang K, Chi H, et al. Treatment effect estimation with adjustment feature selection[C]//Proceedings of the 29th ACM SIGKDD Conference on Knowledge Discovery and Data Mining. 2023: 2290-2301.

**Questions:**

See weakness

---

### Official Review · Reviewer_wUiz · 2023-10-31

**Soundness:** 3 good
**Presentation:** 2 fair
**Contribution:** 2 fair
**Rating:** 5
**Confidence:** 3

**Summary:**

The paper proposes a method CiVAE to control for post-treatment bias. The method leverages three non-verifiable assumptions and under these assumptions, authors can separate the confounding and post-treatment bias. The authors validate the proposed method under synthetic experiments.

**Strengths:**

The paper focuses on a important problem to mitigating the post-treatment bias. Authors clearly state the problem and propose a method that can alleviate the bias under required assumptions.

**Weaknesses:**

The motivating example seems unrealistic and hard to follow. 1) why outcome is age, why would we be interested in estimating the causal effect of age if switching a job from offline to online (and why would there be any causal effect), I would expect age as an exogenous variable; 2) in many companies, the seniority of a job should be easy to quantify (e.g., job title, years since joined), the motivating example is not very realistic in this sense. The authors should pick a different example.

The assumptions in the paper are non-verifiable and hard to justify in practice. I cannot see how practitioners can reason that assumptions 1-3 are satisfied in their specific setup with domain knowledge only, which limits the practical value of the proposed method. Can authors provide any justification of when these assumptions would hold in practice and how can one verify them?

On a similar note, since such assumptions may not hold and cannot be verified in general, I would expect authors show the potential drawback of the proposed method when such assumptions not hold in the data. Will it create additional harm to the ATE estimation or can the authors show such harm can be bounded, or bounded by how much? Since ATE estimation usually corresponds to important policy questions, answering such questions is important.

Definition: DCEV spelled wrong in Def 1.

**Questions:**

See weakness.